# Gradient flow in the gaussian covariate model: exact solution of learning curves and multiple descent structures

## Abstract

A recent line of work has shown remarkable behaviors of the generalization error curves in simple learning models. Even the least-squares regression has shown atypical features such as the model-wise double descent, and further works have observed triple or multiple descents. Another important characteristic are the epoch-wise descent structures which emerge during training. The observations of model-wise and epoch-wise descents have been analytically derived in limited theoretical settings (such as the random feature model) and are otherwise experimental. In this work, we provide a full and unified analysis of the whole time-evolution of the generalization curve, in the asymptotic large-dimensional regime and under gradient-flow, within a wider theoretical setting stemming from a gaussian covariate model. In particular, we cover most cases already disparately observed in the literature, and also provide examples of the existence of multiple descent structures as a function of a model parameter or time. Furthermore, we show that our theoretical predictions adequately match the learning curves obtained by gradient descent over realistic datasets. Technically we compute averages of rational expressions involving random matrices using recent developments in random matrix theory based on "linear pencils". Another contribution, which is also of independent interest in random matrix theory, is a new derivation of related fixed point equations (and an extension there-off) using Dyson brownian motions.

## 1 Introduction

### 1.1 Preliminaries

With growing computational resources, it has become customary for machine learning models to use a *huge* number of parameters (billions of parameters in Brown et al. (2020)), and the need for scaling laws has become of utmost importance Hoffmann et al. (2022). Therefore it is of great relevance to study the asymptotic (or "thermodynamic") limit of simple models in which the number of parameters and data samples are sent to infinity. A landmark progress made by considering these theoretical limits, is the analytical (oftentimes rigorous) calculation of precise double-descent curves for the generalization error starting with Belkin et al. (2020); Hastie et al. (2019); Mei & Montanari (2019), Advani et al. (2020), d'Ascoli et al. (2020), Gerace et al. (2020), Deng et al. (2021), Kini & Thrampoulidis (2020) confirming in a precise (albeit limited) theoretical setting the experimental phenomenon initially observed in Belkin et al. (2019), Geiger et al. (2019); Spigler et al. (2019), Nakkiran et al. (2020a). Further derivations of triple or even multiple descents for the generalization error have also been performed d'Ascoli et al. (2020); Nakkiran et al. (2020b); Chen et al. (2021); Richards et al. (2021); Wu & Xu (2020). Other aspects of multiples descents have been explored in Lin & Dobriban (2021); Adlam & Pennington (2020b) also for the Neural tangent kernel in Adlam & Pennington (2020a). The tools in use come from modern random matrix theory Pennington & Worah (2017); Rashidi Far et al. (2006); Mingo & Speicher (2017), and statistical physics methods such as the replica method Engel & Van den Broeck (2001).

In this paper we are concerned with a line of research dedicated to the precise *time-evolution* of the generalization error under gradient flow corroborating, among other things, the presence of epoch-wise descents structures Crisanti & Sompolinsky (2018); Bodin & Macris (2021) observed in

Nakkiran et al. (2020a). We consider the gradient flow dynamics for the training and generalisation errors in the setting of a Gaussian Covariate model, and develop analytical methods to track the whole time evolution. In particular, for infinite times we get back the predictions of the *least square estimator* which have been thoroughly described in a similar model by Loureiro et al. (2021).

In the next paragraphs we set-up the model together with a list of special realizations, and describe our main contributions.

## 1.2 MODEL DESCRIPTION

**Generative Data Model:**    In this paper, we use the so-called Gaussian Covariate model in a teacher-student setting. An observation in our data model is defined through the realization of a gaussian vector $z \sim \mathcal{N}(0, \frac{1}{d}I_d)$. The teacher and the student obtain their observations (or two different views of the world) with the vectors $x \in \mathbb{R}^{p_B}$ and $\hat{x} \in \mathbb{R}^{p_A}$ respectively, which are given by the application of two linear operations on $z$. In other words there exists two matrices $B \in \mathbb{R}^{d \times p_B}$ and $A \in \mathbb{R}^{d \times p_A}$ such that $x = B^T z$ and $\hat{x} = A^T z$. Note that the generated data can also be seen as the output of a generative 1-layer linear network. In the following, the structure of $A$ and $B$ is pretty general as long as it remains independent of the realization $z$: the matrices may be random matrices or block-matrices of different natures and structures to capture more sophisticated models. While the models we treat are defined through appropriate $A$ and $B$, we will often only need the structure of $U = AA^T$ and $V = BB^T$.

A direct connection can be made with the Gaussian Covariate model described in Loureiro et al. (2021) which suggests considering directly observations $\bar{x} = (x^T, \hat{x}^T)^T \sim \mathcal{N}(0, \Sigma)$ for a given covariance structure $\Sigma$. The spectral theorem provides the existence of orthonormal matrix $O$ and diagonal $D$ such that $\Sigma = O^T DO$ and $D$ contains $d$ non-zero eigenvalues in a squared block $D_1$ and $p_A + p_B - d$ zero eigenvalues. We can write $D = J^T D_1 J$ with $J = (I_d|0_{p_A+p_B-d})$. Therefore if we let $z = \frac{1}{\sqrt{d}}D_1^{-\frac{1}{2}}JO\bar{x}$ which has variance $\frac{1}{d}I_d$, then upon noticing $JJ^T = I_d$ and defining $(A|B)^T = \sqrt{d}O^T J^T D_1^{\frac{1}{2}}$ we find $(A|B)^T z \sim \mathcal{N}(0, \Sigma)$.

The Gaussian Covariate model unifies many different models as shown in Table 1. These special cases are all discussed in section 3 and Appendix D

Table 1: Different matrices and corresponding models

| Target Matrix $B$ | Estimator Matrix $A$ | Corresponding Model |
|---|---|---|
| $\begin{pmatrix} r\sqrt{\frac{d}{p}}I_p & 0 \\ 0 & \sigma\sqrt{\frac{d}{q}}I_q \end{pmatrix}$ | $\begin{pmatrix} \sqrt{\frac{d}{p}}I_p \\ O_{q\times p} \end{pmatrix}$ | Ridgeless regression with signal $r$ and noise $\sigma$ |
| $\begin{pmatrix} \sqrt{\frac{r^2 d}{p}}I_{\gamma p} & 0 & 0 \\ 0 & \sqrt{\frac{r^2 d}{p}}I_{\gamma' p} & 0 \\ 0 & 0 & \sqrt{\frac{\sigma^2 d}{q}}I_q \end{pmatrix}$ | $\begin{pmatrix} \sqrt{\frac{d}{\gamma p}}I_{\gamma p} \\ O_{(1-\gamma)p \times \gamma d} \\ O_{q \times \gamma d} \end{pmatrix}$ | Mismatched ridgeless regression withz signal $r$ and noise $\sigma$ and mismatch parameter $\gamma$ with $\gamma + \gamma' = 1$ |
| $\begin{pmatrix} I_{\gamma d} & 0 & 0 & 0 \\ 0 & I_{\gamma d} & 0 & 0 \\ 0 & 0 & \ddots & \vdots \\ 0 & 0 & \cdots & I_{\gamma d} \end{pmatrix}$ | $\begin{pmatrix} \frac{1}{\alpha^0}I_{\gamma d} & 0 & 0 \\ 0 & \ddots & \vdots \\ 0 & \cdots & \frac{1}{\alpha^{\frac{p-1}{2}}}I_{\gamma d} \end{pmatrix}$ | non-isotropic ridgless regression noiseless with a $\alpha$ polynomial distorsion of the inputs scalings |
| $\begin{pmatrix} r\sqrt{\frac{d}{p}}I_p & O_{p\times q} \\ O_{N\times p} & O_{N\times q} \\ O_{q\times p} & \sigma\sqrt{\frac{d}{q}}I_q \end{pmatrix}$ | $\begin{pmatrix} \mu\sqrt{\frac{d}{p}}W \\ \nu\sqrt{\frac{d}{p}}I_N \\ O_{q\times N} \end{pmatrix}$ | Random features regression of a noisy linear function with $W$ the random weights and $(\mu, \nu)$ describing a non-linear activation function |
| $\begin{pmatrix} \sqrt{\omega_1} & 0 & \cdots & 0 \\ 0 & \sqrt{\omega_2} & \cdots & 0 \\ \vdots & \vdots & \ddots & \vdots \\ 0 & 0 & \cdots & \sqrt{\omega_d} \end{pmatrix}$ | $\begin{pmatrix} \sqrt{\omega_1} & 0 & \cdots & 0 \\ 0 & \sqrt{\omega_2} & \cdots & 0 \\ \vdots & \vdots & \ddots & \vdots \\ 0 & 0 & \cdots & \sqrt{\omega_d} \end{pmatrix}$ | Further Kernel methods |

**Learning task:**    We consider the problem of learning a linear teacher function $f_d(x) = \beta^{*T}x$ with $x$ and $\hat{x}$ sampled as defined above, and with $\beta^* \in \mathbb{R}^p$ a column vectors. This hidden vector $\beta^*$

(to be learned) can potentially be a deterministic vector. We suppose that we have $n$ data-points $(z_i, y_i)_{1 \le i \le n}$ with $x_i = Bz_i, \hat{x}_i = Az_i$. This data can be represented as the $n \times d$ matrix $Z \in \mathbb{R}^{n \times d}$ where $z_i^T$ is the $i$-th row of $Z$, and the column vector vector $Y \in \mathbb{R}^n$ with $i$-th entry $y_i$. Therefore, we have the matrix notation $Y = ZB\beta^*$. We can also set $X = ZB$ so that $Y = X\beta^*$.

In the same spirit, we define the estimator of the student $\hat{y}_\beta(z) = \beta^T x = z^T A\beta$. We note that in general the dimensions of $\beta$ and $\beta^*$ (i.e., $p_A$ and $p_B$) are not necessarily equal as this depends on the matrices $B$ and $A$. We have $\hat{Y} = ZA\beta = \hat{X}\beta$ for $\hat{X} = ZA$.

**Training and test error:**  We will consider the training error $\mathcal{E}_{\text{train}}^\lambda$ and test errors $\mathcal{E}_{\text{gen}}$ with a regularization coefficient $\lambda \in \mathbb{R}_+^*$ defined as

$$\mathcal{E}_{\text{train}}^\lambda(\beta) = \frac{1}{n}\|\hat{Y} - Y\|_2^2 + \frac{\lambda}{n}\|\beta\|_2^2, \quad \mathcal{E}_{\text{gen}}(\beta) = \mathbb{E}_{z \sim \mathcal{N}(0, \frac{I_d}{d})}\left[(z^T A\beta - z^T B\beta^*)^2\right] \quad (1)$$

It is well known that the least-squares estimator $\hat{\beta} = \arg\min \mathcal{H}(\beta)$ is given by the Thikonov regression formula $\hat{\beta}^\lambda = (\hat{X}^T \hat{X} + \lambda I)^{-1} \hat{X}^T Y$ and that in the limit $\lambda \to 0$, this estimator converges towards the $\hat{\beta}^0$ given by the Moore-Penrose inverse $\hat{\beta}^0 = (\hat{X}^T \hat{X})^+ \hat{X}^T Y$.

**Gradient-flow:**  We use the gradient-flow algorithm to explore the evolution of the test error through time with $\frac{\partial \beta_t}{\partial t} = -\frac{n}{2}\nabla_\beta \mathcal{E}_{\text{train}}^\lambda(\beta_t)$. In practice, for numerical calculations we use the discrete-time version, gradient-descent, which is known to converge towards the aforementioned least-squares estimator provided a sufficiently small time-step (in the order of $\frac{1}{\lambda_{\text{max}}}$ where $\lambda_{\text{max}}$ is the maximum eigenvalue of $\hat{X}^T \hat{X}$). The upfront coefficient $n$ on the gradient is used so that the test error scales with the dimension of the model and allows for considering the evolution in the limit $n, d, p_A, p_B \to +\infty$ with a fixed ratios $\frac{n}{d}, \frac{p_A}{d}, \frac{p_B}{d}$. We will note $\phi = \frac{n}{d}$.

### 1.3 Contributions

1. We provide a *general unified framework* covering multiple models in which we derive, in the asymptotic large size regime, the *full time-evolution* under gradient flow dynamics of the training and generalization errors for teacher-student settings. In particular, in the infinite time-limit we check that our equations reduce to those of Loureiro et al. (2021) (as should be expected). But with our results we now have the possibility to explore quantitatively potential advantages of different stopping times: indeed our formalism allows to compute the time derivative of the generalization curve at any point in time.

2. Various special cases are illustrated in section 3, and among these a simpler re-derivation of the whole dynamics of the random features model Bodin & Macris (2021), the full dynamics for kernel methods, and situations exhibiting multiple descent curves both as a function of model parameters and time (See section 3.2 and Appendix D.2). In particular, our analysis allows to design multiple descents with respect to the training epochs.

3. We show that our equations can also capture the learning curves over realistic datasets such as MNIST with gradient descent (See section 3.4 and Appendix D.5), extending further the results of Loureiro et al. (2021) to the time dependence of the curves. This could be an interesting guideline for deriving scaling laws for large learning models.

4. We use modern random matrix techniques, namely an improved version of the linear-pencil method - recently introduced in the machine learning community by Adlam et al. (2019) - to derive asymptotic limits of traces of rational expressions involving random matrices. Furthermore we propose a new derivation an important fixed point equation using Dyson brownian motion which, although non-rigorous, should be of independent interest (See Appendix E).

**Notations:**  We will use $\text{Tr}_d[\cdot] \equiv \lim_{d \to +\infty} \frac{1}{d}\text{Tr}[\cdot]$ and similarly for $\text{Tr}_n[\cdot]$. We also occasionally use $N_d(v) = \lim_{d \to +\infty} \frac{1}{d}\|v\|_2$ for a vector $v$ (when the limit exists).

## 2 Main results

We resort to the high-dimensional assumptions (see Bodin & Macris (2021) for similar assumptions).

**Assumptions 2.1 (High-Dimensional assumptions)** *In the high-dimensional limit, i.e, when $d \rightarrow +\infty$ with all ratios $\frac{n}{d}$, $\frac{p_A}{d}$, $\frac{p_B}{d}$ fixed, we assume the following*

*1. All the traces $Tr_d[\cdot]$, $Tr_n[\cdot]$ concentrate on a deterministic value.*

*2. There exists a sequence of complex contours $\Gamma_d \subset \mathbb{C}$ enclosing the eigenvalues of the random matrix $\hat{X}^T \hat{X} \in \mathbb{R}^{d \times d}$ but not enclosing $-\lambda$, and there exist also a fixed contour $\Gamma$ enclosing the support of the limiting (when $d \rightarrow +\infty$) eigenvalue distribution of $\hat{X}^T \hat{X}$ but not enclosing $-\lambda$.*

With these assumptions in mind, we derive the precise time evolution of the test error in the high-dimensional limit (see result 2.1) and similarly for the training error (see result 2.4). We will also assume that the results are still valid in the case $\lambda = 0$ as suggested in Mei & Montanari (2019).

## 2.1 TIME EVOLUTION FORMULA FOR THE TEST ERROR

**Result 2.1** *The limiting test error time evolution for a random initialization $\beta_0$ such that $N_d(\beta_0) = r_0$ and $\mathbb{E}[\beta_0] = 0$ is given by the following expression:*

$$\bar{\mathcal{E}}_{gen}(t) = c_0 + r_0^2 \mathcal{B}_0(t) + \mathcal{B}_1(t) \tag{2}$$

*with $V^* = B\beta^* \beta^{*T} B^T$ and $c_0 = Tr_d[V^*]$ and:*

$$\mathcal{B}_1(t) = \frac{-1}{4\pi^2} \oint_\Gamma \oint_\Gamma \frac{(1 - e^{-t(x+\lambda)})(1 - e^{-t(y+\lambda)})}{(x+\lambda)(y+\lambda)} f_1(x,y) \mathrm{d}x \mathrm{d}y + \frac{1}{i\pi} \oint_\Gamma \frac{1 - e^{-t(z+\lambda)}}{z+\lambda} f_2(z) \mathrm{d}z \tag{3}$$

$$\mathcal{B}_0(t) = \frac{-1}{2i\pi} \oint_\Gamma e^{-2t(z+\lambda)} f_0(z) \mathrm{d}z \tag{4}$$

*where $f_1(x,y) = f_2(x) + f_2(y) + \tilde{f}_1(x,y) - c_0$ and:*

$$\tilde{f}_1(x,y) = Tr_d \left[ (\phi U + \zeta_x I)^{-1} (\zeta_x \zeta_y V^* + \tilde{f}_1(x,y)\phi U^2)(\phi U + \zeta_y I)^{-1} \right] \tag{5}$$

$$f_2(z) = c_0 - Tr_d \left[ \zeta_z V^* (\phi U + \zeta_z I)^{-1} \right] \tag{6}$$

$$f_0(z) = -\left(1 + \frac{\zeta_z}{z}\right) \tag{7}$$

*and $\zeta_z$ given by the self-consistent equation:*

$$\zeta_z = -z + Tr_d \left[ \zeta_z U (\phi U + \zeta_z I)^{-1} \right] \tag{8}$$

The former result can be expressed in terms of expectations w.r.t the joint limiting eigenvalue distributions of $U$ and $V^*$ when they commute with each other.

**Result 2.2** *Besides, when $U$ and $V^*$ commute, let $u, v^*$ be jointly-distributed according to $U$ and $V^*$ eigenvalues respectively. Then:*

$$\tilde{f}_1(x,y) = \mathbb{E}_{u,v^*} \left[ \frac{\zeta_x \zeta_y v^* + \tilde{f}_1(x,y)\phi u^2}{(\phi u + \zeta_x)(\phi u + \zeta_y)} \right], \quad f_2(z) = c_0 - \mathbb{E}_{u,v^*} \left[ \frac{\zeta_z v^*}{\phi u + \zeta_z} \right] \tag{9}$$

$$\zeta_z = -z + \mathbb{E}_u \left[ \frac{\zeta_z u}{\phi u + \zeta_z} \right] \tag{10}$$

Notice also that in the limit $t \rightarrow \infty$:

$$\mathcal{B}_1(+\infty) = f_1(-\lambda, -\lambda) - 2f_2(-\lambda) = \tilde{f}_1(-\lambda, -\lambda) - c_0, \quad \mathcal{B}_0(+\infty) = 0 \tag{11}$$

which leads to the next result.

**Result 2.3** *In the limit $t \rightarrow \infty$, the limiting test error is given by $\bar{\mathcal{E}}_{gen}(+\infty) = \tilde{f}_1(-\lambda, -\lambda)$.*

**Remark 1**  Notice that the matrix $V^*$ is of rank one depending on the hidden vector $\beta^*$. However, it is also possible to calculate the average generalization (and training) error over a prior distribution $\beta^* \sim \mathcal{P}^*$. Averaging $\mathbb{E}_{\beta^* \sim \mathcal{P}^*}[\bar{\mathcal{E}}_{\text{gen}}]$ propagates the expectation within $\mathbb{E}_{\beta^* \sim \mathcal{P}^*}[\mathcal{B}_0(t)]$ and $\mathbb{E}_{\beta^* \sim \mathcal{P}^*}[\mathcal{B}_1(t)]$, which propagates it further into the traces of $\mathbb{E}_{\beta^* \sim \mathcal{P}^*}[\tilde{f}_1]$ and $\mathbb{E}_{\beta^* \sim \mathcal{P}^*}[f_2]$. In fact we find:

$$\mathbb{E}_{\mathcal{P}^*}[\tilde{f}_1(x,y)] = \text{Tr}_d\left[(\phi U + \zeta_x I)^{-1}(\zeta_x \zeta_y \mathbb{E}_{\mathcal{P}^*}[V^*] + \mathbb{E}_{\mathcal{P}^*}[\tilde{f}_1(x,y)]\phi U^2)(\phi U + \zeta_y I)^{-1}\right] \quad (12)$$

$$\mathbb{E}_{\beta^* \sim \mathcal{P}^*}[f_2(z)] = c_0 - \text{Tr}_d\left[\zeta_z \mathbb{E}_{\mathcal{P}^*}[V^*](\phi U + \zeta_z I)^{-1}\right] \quad (13)$$

In conclusion, we find that $\mathbb{E}_{\beta^* \sim \mathcal{P}^*}[\bar{\mathcal{E}}_{\text{gen}}]$ follows the same equations as $\bar{\mathcal{E}}_{\text{gen}}$ in result 2.1 with $\mathbb{E}_{\beta^* \sim \mathcal{P}^*}[V^*]$ instead of $V^*$. In the following, we will consider $V^*$ without any distinction whether it comes from a specific vector $\beta^*$ or averaged through a sample distribution $\mathcal{P}^*$.

**Remark 2**  In the particular case where $U$ is diagonal, the matrix $V^*$ can be replaced by the following diagonal matrix $\tilde{V}^*$ which, in fact, commutes with $U$:

$$\tilde{V}^* = \begin{pmatrix} [V^*]_{11}[\beta^*]_1^2 & 0 & \dots & 0 \\ 0 & [V^*]_{22}[\beta^*]_2^2 & \dots & 0 \\ \vdots & \vdots & \ddots & \vdots \\ 0 & 0 & \dots & [V^*]_{dd}[\beta^*]_d^2 \end{pmatrix} \quad (14)$$

This comes essentially from the fact that given a diagonal matrix $D$ and a non-diagonal matrix $A$, then $[DA]_{ii} = [D]_{ii}[A]_{ii}$. This is particularly helpful, and shows that in many cases the calculations of $\tilde{f}_1$ or $f_2$ remain tractable even for a deterministic $\beta^*$ (see the example in Appendix D.3) .

**Remark 3**  Sometimes $U = AA^T$ and $V = BB^T$ are more difficult to handle than their dual counterparts $U_\star = \phi A^T A$ and $V_\star = \phi B^T B$ together with the additional matrix $\Xi = \phi A^T B$. The following expressions are thus very useful (See Appendix C):

$$f_1(x,y) = \text{Tr}_n\left[(U_\star + \zeta_x I)^{-1}((\Xi \beta^* \beta^{*T} \Xi^T) + \tilde{f}_1(x,y)U_\star)U_\star(U_\star + \zeta_y I)^{-1}\right] \quad (15)$$

$$f_2(z) = \text{Tr}_n\left[(\Xi \beta^* \beta^{*T} \Xi^T)(U_\star + \zeta_z I)^{-1}\right] \quad (16)$$

$$\zeta_z = -z + \text{Tr}_n\left[\zeta_z U_\star (U_\star + \zeta_z I)^{-1}\right] \quad (17)$$

In fact, when $x = y = -\lambda$ (which corresponds to the limit when $t \to \infty$), these are the same expressions as (59) in Loureiro et al. (2021) with the appropriate change of variable $\lambda(1 + V) \to \zeta$ and $\tilde{f}_1 \to \rho + q - 2m$.

## 2.2  Time evolution formula for the training error

**Result 2.4**  *The limiting training error time evolution is given by the following expression:*

$$\bar{\mathcal{E}}_{train}^0(t) = c_0 + r_0^2 \mathcal{H}_0(t) + \mathcal{H}_1(t) \quad (18)$$

*with:*

$$\mathcal{H}_1(t) = \frac{-1}{4\pi^2} \oint_\Gamma \oint_\Gamma \frac{(1 - e^{-t(x+\lambda)})(1 - e^{-t(y+\lambda)})}{(x+\lambda)(y+\lambda)} h_1(x,y)\mathrm{d}x\mathrm{d}y + \frac{1}{i\pi} \oint_\Gamma \frac{1 - e^{-t(z+\lambda)}}{z+\lambda} h_2(z)\mathrm{d}z \quad (19)$$

$$\mathcal{H}_0(t) = \frac{-1}{2i\pi} \oint_\Gamma e^{-2t(z+\lambda)} h_0(z)\mathrm{d}z \quad (20)$$

*where $h_1(x,y) = h_2(x) + h_2(y) + \tilde{h}_1(x,y) - c_0$ and with $\eta_z = \frac{-z}{\zeta_z}$:*

$$\tilde{h}_1(x,y) = \eta_x \eta_y \tilde{f}_1(x,y), \quad h_2(z) = \eta_z(c_0 f_0(z) + f_2(z)), \quad h_0(z) = \eta_z f_0(z) \quad (21)$$

*Eventually, in the limit $t \to \infty$ we find:*

$$\mathcal{H}_1(+\infty) = h_1(-\lambda, -\lambda) - 2h_2(-\lambda) = \tilde{h}_1(-\lambda, -\lambda) - c_0, \quad \mathcal{H}_0(+\infty) = 0 \quad (22)$$

**Result 2.5** *In the limit $t \to \infty$, we have the relation $\bar{\mathcal{E}}_{train}^0(+\infty) = \eta_{-\lambda}^2 \bar{\mathcal{E}}_{gen}(+\infty)$*

We notice the same proportionality factor $\eta_{-\lambda}^2 = \left( \frac{\lambda}{\zeta(-\lambda)} \right)^2$ as already stated in Loureiro et al. (2021), however interestingly, in the time evolution of the training error, such a factor is not valid as we have $h_2(z) \neq \eta_z f_2(z)$.

## 3 APPLICATIONS AND EXAMPLES

We discuss some of the models provided in table 1 and some others in Appendix D.

### 3.1 RIDGELESS REGRESSION OF A NOISY LINEAR FUNCTION

**Target function** Consider the following noisy linear function $y(x) = r x^T \beta_0^* + \sigma \epsilon$ for some constant $\sigma \in \mathbb{R}^+$ and $\epsilon \sim \mathcal{N}(0, 1)$, and a hidden vector $\beta_0^* \sim \mathcal{N}(0, I_p)$. Assume we have a data matrix $X \in \mathbb{R}^{n \times p}$. In order to incorporate the noise in our structural matrix $B$, we consider an additional parameter $q(d)$ that grows linearly with $d$ and such that $d = p + q$. Let $\phi_0 = \frac{n}{p}$. Therefore $\phi = \frac{n}{d} = \frac{n}{p} \frac{p}{d} = \phi_0 \psi$. Also, we let $\beta^{*T} = (\beta_0^{*T} | \beta_1^T) \sim \mathcal{N}(0, I_{p+q})$ and we consider an average $V^*$ over $\beta^*$. We construct the following block-matrix $B$ and compute the averaged $V^*$ as follow:

$$B = \begin{pmatrix} r\sqrt{\frac{d}{p}} I_p & 0 \\ 0 & \sigma\sqrt{\frac{d}{q}} I_q \end{pmatrix} \implies V^* = \begin{pmatrix} r^2 \frac{1}{\psi} I_p & 0 \\ 0 & \sigma^2 \frac{1}{1-\psi} I_q \end{pmatrix} \tag{23}$$

Now let's consider the random matrix $Z \in \mathbb{R}^{n \times d}$ and split it into two sub-blocks $Z = \left( \sqrt{\frac{p}{d}} X \mid \sqrt{\frac{q}{d}} \Sigma \right)$. The framework of the paper yields the following output vector:

$$Y = ZB\beta^* = rX\beta_0^* + \sigma\xi \tag{24}$$

where $\xi = \Sigma \beta_1^*$ is used as a proxy for the noise $\epsilon$.

**Estimator** Now let's consider the linear estimator $\hat{y}_t = x^T \beta_t$. To capture the structure of this model, we use the following block-matrix $A$ and compute the resulting matrix $U$:

$$A = \begin{pmatrix} \sqrt{\frac{d}{p}} I_p \\ 0_{q \times p} \end{pmatrix} \implies U = \begin{pmatrix} \frac{1}{\psi} I_p & 0 \\ 0 & 0_{q \times q} \end{pmatrix} \tag{25}$$

Therefore, it is straightforward to check that we have indeed: $\hat{Y}_t = ZA\beta_t = X\beta_t$.

**Analytic result** In this specific example, $U$ and $V^*$ obviously commute and the result 2.2 can thus be used. First we derive the joint-distribution of the eigenvalues:

$$\mathcal{P}\left( u = \frac{1}{\psi}, v = \frac{r^2}{\psi} \right) = \psi \qquad \mathcal{P}\left( u = 0, v = \frac{\sigma^2}{1-\psi} \right) = 1 - \psi \tag{26}$$

In this specific example, we focus only on rederiving the high-dimensional generalization error without any regularization term ($\lambda = 0$) for the minimum least-squares estimator. So we calculate $\zeta = \zeta(0)$ as follows: $\zeta = \psi \frac{\zeta \frac{1}{\psi}}{\frac{\phi}{\psi} + \zeta} + 0$ implies $\zeta^2 + \phi_0 \zeta = \zeta$ so $\zeta \in \{0, 1 - \phi_0\}$. For $\tilde{f}_1$ we get:

$$\tilde{f}_1 = \psi \frac{\tilde{f}_1 \frac{\phi}{\psi^2}}{(\frac{\phi}{\psi} + \zeta)^2} + \psi \frac{r^2}{\psi} \frac{\zeta^2}{(\frac{\phi}{\psi} + \zeta)^2} + (1 - \psi) \frac{\sigma^2}{1-\psi} \frac{\zeta^2}{(\zeta)^2} \tag{27}$$

In fact, the expression can be simplified as follow (without the constants $\phi, \psi$):

$$\left( 1 - \frac{\phi_0}{(\phi_0 + \zeta)^2} \right) \tilde{f}_1 = r^2 \frac{\zeta^2}{(\phi_0 + \zeta)^2} + \sigma^2 \tag{28}$$

Using both solutions $\zeta = 0$ or $\zeta = 1 - \phi_0$ yields the same results as in Hastie et al. (2019); Belkin et al. (2020) using 2.3:

$$\mathcal{E}_{gen}(+\infty) = \begin{cases} \sigma^2 \frac{\phi_0}{\phi_0 - 1} & (\zeta = 0) \\ r^2(1 - \phi_0) + \sigma^2 \frac{1}{1 - \phi_0} & (\zeta = 1 - \phi_0) \end{cases} \tag{29}$$

### 3.2 NON-ISOTROPIC RIDGELESS REGRESSION OF A NOISELESS LINEAR MODEL

Non-isotropic models have been studied in Dobriban & Wager (2018) and then also Wu & Xu (2020); Richards et al. (2021); Nakkiran et al. (2020b); Chen et al. (2021) where multiple-descents curve have been observed or engineered. In this section, we extend this idea to show that any number of descents can be generated and derive the precise curve of the generalization error as in Figure 1.

**Target function** We use the standard linear model $y(z) = z^T \beta^*$ for a random $\beta^* \sim \mathcal{N}(0, I_d)$. Therefore, we consider the matrix $B = I_d$ and thus $V^* = I_d$ such that $Y = ZB\beta^* = Z\beta^*$.

**Estimator:** Following the structure provided in table 1, the design a matrix $A$ is a scalar matrix with $p \in \mathbb{N}^*$ sub-spaces of different scales spaced by a polynomial progression $\alpha^{-\frac{1}{2}i}$. In other words, the student is trained on a dataset with different scalings. We thus have $U = A^2$ and $\hat{Y}_t = ZA\beta_t$.

**Analytic results** We refer the reader to the Appendix D.2 for the calculation. Depending if $\phi$ is above or below 1, $\zeta$ is the solution of the following equations: $\zeta = 0$ or $1 = \frac{1}{p} \sum_{i=0}^{p-1} \frac{1}{\phi + \alpha^i \zeta}$. In the over-parameterized regime ($\phi < 1$), the generalisation error is fully characterized by the equation:

$$\bar{\mathcal{E}}_{\text{gen}}(+\infty) = \phi(1 - \phi) \left( \frac{1}{p} \sum_{i=0}^{p-1} \frac{\alpha^i \zeta}{(\phi + \alpha^i \zeta)^2} \right)^{-1} - \phi \tag{30}$$

In the asymptotic limit $\alpha \to \infty$, $\zeta$ can be approximated and thus we can derive an asymptotic expansion of $\bar{\mathcal{E}}_{\text{gen}}(+\infty)$ for $\phi \in [0, 1] \setminus \frac{k}{p}\mathbb{Z}$ where clearly, the multiple descents appear as roots of the denominator of the sum:

$$\bar{\mathcal{E}}_{\text{gen}}(+\infty) = \frac{1}{p} \sum_{k=0}^{p-1} \frac{\phi(1 - \phi)}{\left(\phi - \frac{k}{p}\right)\left(\frac{k+1}{p} - \phi\right)} \mathbb{1}_{]\frac{k}{p}; \frac{k+1}{p}[}(\phi) - \phi + o_\alpha(1) \tag{31}$$

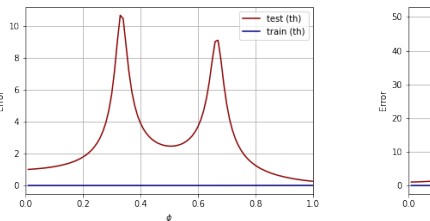
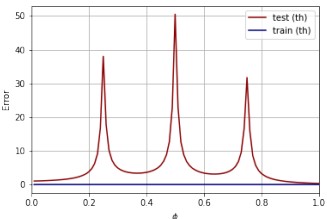

Figure 1: Example of theoretical multiple descents in the least-squares solution for the non-isotropic ridgeless regression model with $p = 3, \lambda = 10^{-7}$ (left) and $p = 4, \lambda = 10^{-13}$ (right), and $\alpha = 10^4$ in both of them.

Interestingly, we can see how these peaks are being formed with the time-evolution of the gradient flow as in Figure 2 with one peak close to $\phi = \frac{1}{3}$ and the second one at $\phi = \frac{2}{3}$. (Note that small $\lambda$ requires more computational resources to have finer resolution at long times, hence here the second peak develops fully after $t = 10^4$). It is worth noticing also the existence of multiple time-descent, in particular at $\phi = 1$ with some "ripples" that can be observed even in the training error.

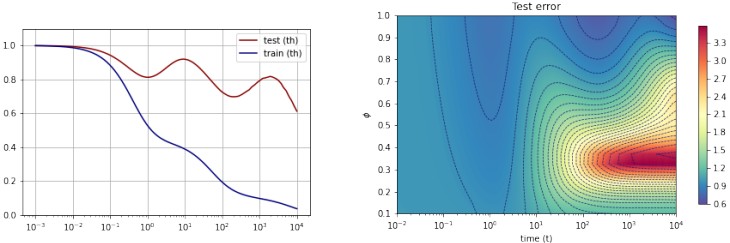

Figure 2: Example of theoretical multiple descents evolution in the non-isotropic ridgeless regression model with $p = 3, \lambda = 10^{-5}, \alpha = 100$ with $\phi = 1$ on the left and a range $\phi \in (0, 1)$ on the right heatmap.

The eigenvalue distribution (See Appendix D.2.1) provides some insights on the existence of these phenomena. As seen in Figure 3, the emergence of a spike is related to the rise of a new "bulk" of eigenvalues, which can be clearly seen around $\phi = \frac{1}{3}$ and $\phi = \frac{2}{3}$ here. Note that there is some analogy for the generic double-descent phenomena described in Hastie et al. (2019) where instead of two bulks, there is a mass in 0 which is arising. Furthermore, the existence of multiple bulks allow for multiple evolution at different scales (with the $e^{-(z+\lambda)t}$ terms) and thus enable the emergence of multiple epoch-wise peaks.

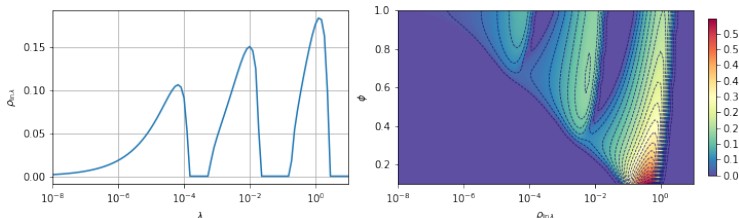

Figure 3: Theoretical (log-)eigenvalue distribution in the non-isotropic ridgeless regression model with $p = 3, \lambda = 10^{-5}, \alpha = 100$ with $\phi = 1$ on the left and a range $\phi \in (0, 1)$ on the right heatmap.

## 3.3 RANDOM FEATURES REGRESSION

In this section, we show that we can derive the learning curves for the random features model introduced in Rahimi & Recht (2008), and we consider the setting described in Bodin & Macris (2021). In this setting, we define the random weight-matrix $W \in \mathbb{R}^{p \times N}$ where $\psi_0 = \frac{N}{p}$ such that $W_{ij} \sim \mathcal{N}(0, \frac{1}{p})$ and $d = p + N + q$ and $\phi = \frac{n}{d}$, $\psi = \frac{p}{d}$, and $\phi_0 = \frac{n}{p} = \frac{n}{d}\frac{d}{p} = \frac{\phi}{\psi}$ (thus $\frac{q}{d} = 1 - (1 + \psi_0)\psi$). So with $Z = \left( \sqrt{\frac{p}{d}}X | \sqrt{\frac{p}{d}}\Omega | \sqrt{\frac{q}{d}}\xi \right)$, using the structures $A$ and $B$ from table 1 we have: $ZA = \mu XW + \nu \Omega$ and $ZB = X + \sigma \xi$, hence the model:

$$\hat{Y} = ZA\beta = (\mu XW + \nu \Omega)\beta \tag{32}$$
$$Y = ZB\beta^* = X\beta_0^* + \sigma \xi \beta_1^* \tag{33}$$

With further calculation that can be found in Appendix D.4, a similar complete time derivation of the random feature regression can be performed with a much smaller linear-pencil than the one suggested in Bodin & Macris (2021). As stated in this former work, the curves derived from this formula track the same training and test error in the high-dimensional limit as the model with the point-wise application of a centered non-linear activation function $f \in L^2(e^{-\frac{x^2}{2}}\mathrm{d}x)$ with $\hat{Y} = \frac{1}{\sqrt{p}}f(\sqrt{p}XW)\beta$. More precisely, with the inner-product defined such that for any function $g \in L^2(e^{-\frac{x^2}{2}}\mathrm{d}x)$, $\langle f, g \rangle = \mathbb{E}_{x \sim \mathcal{N}(0,1)}[f(x)g(x)]$, we derive the equivalent model parameters $(\mu, \nu)$ with $\mu = \langle f, H_{e_1} \rangle$, $\nu^2 = \langle f, f \rangle - \mu^2$ while having the centering condition $\langle f, H_{e_0} \rangle = 0$ where $(H_{e_n})$ is the Hermite polynomial basis.

This transformation is dubbed the Gaussian equivalence principle and has been observed and rigorously proved under weaker conditions in Pennington & Worah (2017); Péché (2019); Hu & Lu (2022), and since then has been applied more broadly for instance in Adlam & Pennington (2020a).

## 3.4 TOWARDS REALISTIC DATASETS

As stated in Loureiro et al. (2021), the training and test error of realistic datasets can also be captured. In this example we track the MNIST dataset and focus on learning the parity of the images ($y = +1$ for even numbers and $y = -1$ for odd-numbers). We refer to Appendix D.5 for thorough discussions of Figures 4 and 5 as well as technical details to obtain them, and other examples. Besides the learning curve profile at $t = +\infty$, the full theoretical time evolution is predicted and matches the experimental runs. In particular, the rise of the double-descent phenomenon is observed through time.

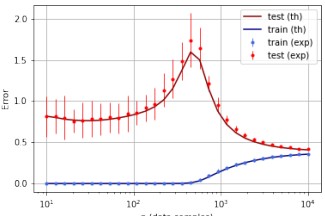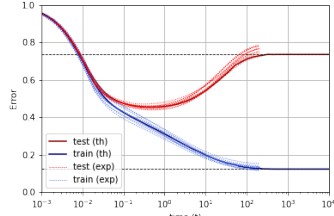

Figure 4: Comparison between the analytical and experimental learning profiles for the minimum least-squares estimator at $\lambda = 10^{-3}$ on the left (20 runs) and the time evolution at $\lambda = 10^{-2}, n = 700$ on the right (10 runs).

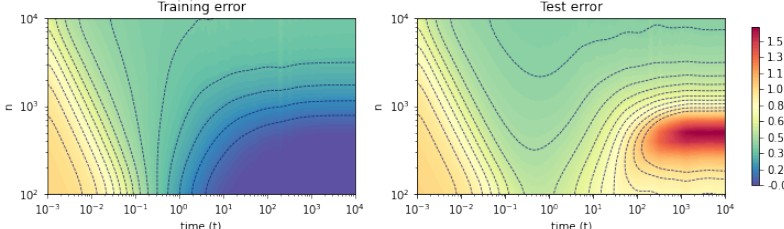

Figure 5: Analytical training error and test error heat-maps for the theoretical gradient flow for $\lambda = 10^{-3}$.

## 4 CONCLUSION

The time-evolution can also be investigated using the dynamical mean field theory (DMFT) from statistical mechanics. We refer the reader to the book Parisi et al. (2020) and a series of recent works Sompolinsky et al. (1988); Crisanti & Sompolinsky (2018); Agoritsas et al. (2018); Mignacco et al. (2020; 2021) for an overview of this tool. This method is a priori unrelated to ours and yields a set of non-linear integro-differential equations for time correlation functions which are in general not solvable analytically and one has to resort to a numerical solution. It would be interesting to understand if for the present model the DMFT equations can be reduced to our set of algebraic equations. We believe it can be a fruitful endeavor to compare in detail the two approaches: the one based on DMFT and the one based on random matrix theory tools and Cauchy integration formulas.

Another interesting direction which came to our knowledge recently is the one taken in Lu & Yau (2022); Hu & Lu (2022) and in Misiakiewicz (2022); Xiao & Pennington (2022), who study the high-dimensional polynomial regime where $n \propto d^{\kappa}$ for a fixed $\kappa$. In particular, it is becoming notorious that changing the scaling can yield additional descents. This regime is out of the scope of the present work but it would be desirable to explore if the linear-pencils and the random matrix tools that we extensively use in this work can extend to these cases.

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

## A  Gradient flow calculations

In this section, we derive the main equations for the gradient flow algorithm, and derive and set of Cauchy integration formula involving the limiting traces of large matrices. The calculation factoring out $Z$ in the limit $d \to \infty$ is pursued in the next section. First, we recall and expand the training error function in 1:

$$\mathcal{E}_{\text{train}}^{\lambda}(\beta_t) = \frac{1}{n}\|Y - \hat{X}\beta_t\|_2^2 + \frac{\lambda}{n}\|\beta_t\|_2^2 \tag{34}$$

$$= \frac{1}{n}\|Y\|_2^2 - \frac{2}{n}Y^T\hat{X}^T\hat{X}\beta_t + \frac{\lambda}{n}\|\beta_t\|_2^2 \tag{35}$$

$$= \frac{1}{n}\|ZB\beta^*\|_2^2 - \frac{2}{n}\beta^{*T}B^TZ^TZA\beta_t + \frac{1}{n}\beta_t^TA^TZ^TZA\beta_t + \frac{\lambda}{n}\|\beta_t\|_2^2 \tag{36}$$

Let $K = (\hat{X}^T\hat{X} + \lambda I)^{-1} = (A^TZ^TZA + \lambda I)^{-1}$ which is invertible for $\lambda > 0$. Therefore, we can write the gradient of the training error for any $\beta$ as:

$$\frac{n}{2}\nabla_\beta\mathcal{E}_{\text{train}}(\beta) = \hat{X}^T(\hat{X}\beta - Y) + \lambda\beta = (\hat{X}^T\hat{X} + \lambda I)\beta - \hat{X}^TY = K^{-1}\beta - \hat{X}^TY \tag{37}$$

The gradient flow equations reduces to a first order ODE

$$\frac{\partial\beta_t}{\partial t} = -\frac{n}{2}\nabla_\beta\mathcal{E}_{\text{train}}^{\lambda}(\beta_t) = \tilde{X}^TY - K^{-1}\beta_t \tag{38}$$

The solution can be completely expressed using $L_t = (I - \exp(-tK^{-1}))$ as

$$\beta_t = \exp(-tK^{-1})\beta_0 + (I - \exp(-tK^{-1}))K\tilde{X}^TY \tag{39}$$

$$= (I - L_t)\beta_0 + L_tK\tilde{X}^TX\beta^* \tag{40}$$

In the following two subsections, we will focus on deriving an expression of the time evolution of the test error and training error using these equations averaged over the a centered random vector $\beta_0$ such that $r_0^2 = N_d(\beta_0)^2$.

### A.1  Test error

As above, the test error can be expanded using the fact that on $\mathcal{N}_0 = \mathcal{N}(0, \frac{1}{d})$, we have the identity $\mathbb{E}_{z \sim \mathcal{N}_0}[zz^T] = \frac{1}{d}I_d$:

$$\mathcal{E}_{\text{gen}}(\beta_t) = \mathbb{E}_{z \sim \mathcal{N}_0}\left[(z^TA\beta_t - z^TB\beta^*)^2\right] \tag{41}$$

$$= (A\beta_t - B\beta^*)^T\mathbb{E}_{z \sim \mathcal{N}_0}[zz^T](A\beta_t - B\beta^*) \tag{42}$$

$$= \frac{1}{d}\beta_t^TU\beta_t - \frac{2}{d}\beta^{*T}B^TA\beta_t + \frac{1}{d}\beta^{*T}B^TB\beta^* \tag{43}$$

So expanding the first term yields

$$\beta_t^TU\beta_t = (\beta_0^T(I - L_t) + \beta^{*T}X^T\tilde{X}KL_t)U((I - L_t)\beta_0 + L_tK\tilde{X}^TX\beta^*) \tag{44}$$

$$= \beta_0^T(I - L_t)U(I - L_t)\beta_0 \tag{45}$$

$$+ \beta^{*T}(B^TZ^TZA)KL_tUL_tK(AZ^TZB)\beta^* \tag{46}$$

$$+ 2\beta_0^T(I - L_t)UL_tK(AZ^TZB)\beta^* \tag{47}$$

while the second term yields

$$\beta^{*T}B^TA\beta_t = \beta^{*T}B^TA((I - L_t)\beta_0 + L_tK\tilde{X}^TX\beta^*) \tag{48}$$

$$= \beta^{*T}B^TA(I - L_t)\beta_0 + \beta^{*T}L_tK(A^TZ^TZB)\beta^* \tag{49}$$

Let's consider now the high-dimensional limit $\bar{\mathcal{E}}_{\text{gen}}(t) = \lim_{d \to +\infty}\mathcal{E}_{\text{gen}}(\beta_t)$. We further make the underlying assumption that the generalisation error concentrates on its mean with $\beta_0$, that is to say: $\bar{\mathcal{E}}_{\text{gen}}(t) = \lim_{d \to +\infty}\mathbb{E}_{\beta_0}[\mathcal{E}_{\text{gen}}(\beta_t)]$. Let $V^* = B\beta^*\beta^{*T}B^T$ and $c_0 = \text{Tr}_d[V^*]$, then using the former expanded terms in 41 we find the expression

$$\bar{\mathcal{E}}_{\text{gen}}(t) = c_0 + r_0^2\text{Tr}_d\left[A^T(I - L_t)^2A\right] \tag{50}$$

$$+ \text{Tr}_d\left[Z^TZAKL_tUL_tKA^TZ^TZV^*\right] - 2\text{Tr}_d\left[AL_tKA^TZ^TZV^*\right] \tag{51}$$

So $\bar{\mathcal{E}}_{\text{gen}}(t) = c_0 + r_0^2 \mathcal{B}_0(t) + \mathcal{B}_1(t)$ with:

$$\mathcal{B}_0(t) = \text{Tr}_d\left[A^T(I - L_t)^2 A\right] \tag{52}$$

$$\mathcal{B}_1(t) = \text{Tr}_d\left[Z^T Z A K L_t U L_t K A^T Z^T Z V^*\right] - 2\text{Tr}_d\left[A L_t K A^T Z^T Z V^*\right] \tag{53}$$

Let $K(z) = (\tilde{X}^T \tilde{X} - zI)^{-1}$ the resolvent of $\tilde{X}^T \tilde{X}$, and let's have the convention $K = K(-\lambda)$ to remain consistent with the previous formula. Then for any holomorphic functional $f : \mathbb{U} \to \mathbb{C}$ defined on an open set $\mathbb{U}$ which contains the spectrum of $\tilde{X}^T \tilde{X}$, with $\Gamma$ a contour in $\mathbb{C}$ enclosing the spectrum of $\tilde{X}^T \tilde{X}$ but not the poles of $f$, we have with the extension of $f$ onto $\mathbb{C}^{n \times n}$: $f(\tilde{X}^T \tilde{X}) = \frac{-1}{2i\pi} \oint_\Gamma f(z) K(z) \mathrm{d}z$. For instance, we can apply it for the following expression:

$$KL_t = L_t K = (I - \exp(-t\tilde{X}^T \tilde{X} + t\lambda I))(\tilde{X}^T \tilde{X} - \lambda I)^{-1} \tag{54}$$

$$= \frac{-1}{2i\pi} \oint_\Gamma \frac{1 - e^{-t(z+\lambda)}}{z + \lambda}(\tilde{X}^T \tilde{X} - zI)^{-1}\mathrm{d}z \tag{55}$$

$$= \frac{-1}{2i\pi} \oint_\Gamma \frac{1 - e^{-t(z+\lambda)}}{z + \lambda}K(z)\mathrm{d}z \tag{56}$$

So we can generalize this idea to each trace and rewrite $\mathcal{B}_1(t)$ and $\mathcal{B}_0(t)$ with

$$\mathcal{B}_1(t) = \frac{-1}{4\pi^2} \oint_\Gamma \oint_\Gamma \frac{(1 - e^{-t(x+\lambda)})(1 - e^{-t(y+\lambda)})}{(x+\lambda)(y+\lambda)}f_1(x,y)\mathrm{d}x\mathrm{d}y + \frac{1}{i\pi} \oint_\Gamma \frac{1 - e^{-t(z+\lambda)}}{z + \lambda}f_2(z)\mathrm{d}z \tag{57}$$

$$\mathcal{B}_0(t) = \frac{-1}{2i\pi} \oint_\Gamma e^{-2t(z+\lambda)}f_0(z)\mathrm{d}z \tag{58}$$

where we introduce the set of functions $f_1(x,y)$, $f_2(z)$ and $f_0(z)$

$$f_1(x,y) = \text{Tr}_d\left[Z^T Z A K(x) A A^T K(y) A^T Z^T Z V^*\right] \tag{59}$$

$$f_2(z) = \text{Tr}_d\left[A K(z) A^T Z^T Z V^*\right] \tag{60}$$

$$f_0(z) = \text{Tr}_d\left[A K(z) A^T\right] \tag{61}$$

Let $G(x) = (U Z^T Z - xI)^{-1}$, using the push-through identity, it is straightforward that $A K(z) A = G(z) U = U G(z)^T$. This help us reduce further the expression of $f_1$ into smaller terms which will be easier to handle with linear-pencils later on

$$f_1(x,y) = \text{Tr}_d\left[Z^T Z U G(x)^T G(y) U Z^T Z V^*\right] \tag{62}$$

$$= \text{Tr}_d\left[(G(x)^{-1} + xI)^T G(x)^T G(y)(G(y)^{-1} + yI)V^*\right] \tag{63}$$

$$= \text{Tr}_d\left[(I + yG(y))V^*(I + xG(x))^T\right] \tag{64}$$

$$= c_0 + y\text{Tr}_d\left[G(y)V^*\right] + x\text{Tr}_d\left[G(x)V^*\right] + xy\text{Tr}_d\left[G(x)V^*G(y)^T\right] \tag{65}$$

Similarly with $f_2$ and $f_0$, they can be rewritten as

$$f_2(z) = \text{Tr}_d\left[G(z)U Z^T Z V^*\right] \tag{66}$$

$$= \text{Tr}_d\left[G(z)(G(z)^{-1} + zI)V^*\right] \tag{67}$$

$$= c_0 + z\text{Tr}_d\left[G(z)V^*\right] \tag{68}$$

$$f_0(z) = \text{Tr}_d\left[G(z)U\right] \tag{69}$$

Hence in fact the definition $\tilde{f}_1(x,y) = xy\text{Tr}_d\left[G(x)V^*G(y)^T\right]$ such that

$$f_1(x,y) = f_2(x) + f_2(y) + \tilde{f}_1(x,y) - c_0 \tag{70}$$

At this point, the equations provided by 57 are valid for any realization $Z$ in the limit $d \to \infty$. We will see in the next section how to simplify these terms by factoring out $Z$.

## A.2 TRAINING ERROR

Similar formulas can be derived for the training error. For the sake of simplicity, we provide a formula to track the training error without the regularization term, that is to say $\mathcal{E}_{\text{train}}^{0}(\beta_t)$ (as in Loureiro et al. (2021)) while still minimizing the loss $\mathcal{E}_{\text{train}}^{\lambda}(\beta_t)$. So using the expanded expression 34, and considering the high-dimensional assumption with concentration $\bar{\mathcal{E}}_{\text{train}}^{0}(t) := \lim_{d\to+\infty} \mathcal{E}_{\text{train}}(\beta_t) = \lim_{d\to+\infty} \mathbb{E}_{\beta_0}[\mathcal{E}_{\text{train}}(\beta_t)]$ we have

$$\bar{\mathcal{E}}_{\text{train}}^{0}(t) = \text{Tr}_n\left[Z^T ZV^*\right] + r_0^2 \text{Tr}_n\left[A^T Z^T ZA(I - L_t)^2\right] \tag{71}$$

$$+ \text{Tr}_n\left[Z^T ZAKL_t A^T Z^T ZAL_t KA^T Z^T ZV^*\right] \tag{72}$$

$$- 2\text{Tr}_n\left[Z^T ZAL_t KA^T Z^T ZV^*\right] \tag{73}$$

First of all, standard random matrix results (for instance see Rubio & Mestre (2011)) assert the result $\text{Tr}_d\left[Z^T ZV^*\right] = \text{Tr}_d\left[Z^T Z\right]\text{Tr}_d\left[V^*\right] = \phi c_0$. This result can also be derived under our random matrix theory framework, for completeness we provide this calculation in C.2. Therefore, we can define $\mathcal{H}_0(t)$ and $\mathcal{H}_1(t)$ such that

$$\bar{\mathcal{E}}_{\text{train}}^{0}(t) = c_0 + r_0^2 \mathcal{H}_0(t) + \mathcal{H}_1(t) \tag{74}$$

where we have the traces

$$\mathcal{H}_0(t) = \text{Tr}_n\left[A^T Z^T ZA(I - L_t)^2\right] \tag{75}$$

$$\mathcal{H}_1(t) = \text{Tr}_n\left[Z^T ZAKL_t(A^T Z^T ZA)L_t KA^T Z^T ZV^*\right] - 2\text{Tr}_n\left[Z^T ZAL_t KA^T Z^T ZV^*\right] \tag{76}$$

And using the functional calculus argument with Cauchy integration formula over the same contour $\Gamma$ we find

$$\mathcal{H}_1(t) = \frac{-1}{4\pi^2}\oint_\Gamma\oint_\Gamma \frac{(1 - e^{-t(x+\lambda)})(1 - e^{-t(x+\lambda)})}{(x+\lambda)(y+\lambda)} h_1(x,y)\mathrm{d}x\mathrm{d}y + \frac{1}{i\pi}\oint_\Gamma \frac{(1 - e^{-t(z+\lambda)})}{(z+\lambda)} h_2(z)\mathrm{d}z \tag{77}$$

$$\mathcal{H}_0(t) = \frac{-1}{2i\pi}\oint_\Gamma e^{-2t(z+\lambda)} h_0(z)\mathrm{d}z \tag{78}$$

Where we use the traces (which only contain algebraic expression of matrices):

$$h_1(x,y) = \text{Tr}_n\left[Z^T ZAK(x)AZ^T ZA^T K(y)A^T Z^T ZV^*\right] \tag{79}$$

$$h_2(z) = \text{Tr}_n\left[Z^T ZAK(z)A^T Z^T ZV^*\right] \tag{80}$$

$$h_0(z) = \text{Tr}_n\left[Z^T ZA^T K(z)A^T\right] \tag{81}$$

The expression of $h_1$ can be reduced to smaller terms as before with $f_1$

$$\phi h_1(x,y) = \text{Tr}_d\left[Z^T ZUG(x)^T Z^T ZG(y)UZ^T ZV^*\right] \tag{82}$$

$$= \text{Tr}_d\left[(G(x)^{-1} + xI)^T G(x)^T Z^T ZG(y)(G(y)^{-1} + yI)V^*\right] \tag{83}$$

$$= \text{Tr}_d\left[Z^T ZV^*\right] + x\text{Tr}_d\left[G(x)^T Z^T ZV^*\right] + y\text{Tr}_d\left[Z^T ZG(y)V^*\right] \tag{84}$$

$$+ xy\text{Tr}_d\left[Z^T ZG(y)V^* G(x)^T\right] \tag{85}$$

$$= c_0\phi + x\text{Tr}_d\left[Z^T ZG(x)V^*\right] + y\text{Tr}_d\left[Z^T ZG(y)V^*\right] \tag{86}$$

$$+ xy\text{Tr}_d\left[Z^T ZG(y)V^* G(x)^T\right] \tag{87}$$

and similarly with $h_2$

$$\phi h_2(z) = \text{Tr}_d\left[Z^T ZG(z)UZ^T ZV^*\right] \tag{88}$$

$$= \text{Tr}_d\left[Z^T ZG(z)(G(z)^{-1} + zI)V^*\right] \tag{89}$$

$$= \text{Tr}_d\left[Z^T ZV^*\right] + z\text{Tr}_d\left[Z^T ZG(z)V^*\right] \tag{90}$$

$$= c_0\phi + z\text{Tr}_d\left[Z^T ZG(z)V^*\right] \tag{91}$$

and similarly with $h_0$

$$\phi h_0(z) = \text{Tr}_d\left[Z^T ZG(z)U\right] \tag{92}$$

$$= \text{Tr}_d\left[G(z)(G(z)^{-1} + zI)\right] \tag{93}$$

$$= 1 + z\text{Tr}_d\left[G(z)\right] \tag{94}$$

We can also define the term $\tilde{h}_1(x,y) = xy\text{Tr}_n\left[ZG(y)V^* G(x)^T Z^T\right]$ so that:

$$h_1(x,y) = h_2(x) + h_2(y) + \tilde{h}_1(x,y) - c_0 \tag{95}$$

## B    TEST ERROR AND TRAINING ERROR LIMITS WITH LINEAR PENCILS

In this section we compute a set of self-consistent equation to derive the high-dimensional evolution of the training and test error. We refer to Appendix E for the definition and result statements concerning the linear pencils.

We will derive essentially two linear-pencils of size $6 \times 6$ and $4 \times 4$ which will enable us to calculate the limiting values for $\tilde{f}_1, f_2, f_0$ for the test error, and $\tilde{h}_1, h_2, h_0$ for the training error. Note that these block-matrices are derived essentially by observing the recursive application of the block-matrix inversion formula and manipulating it so as to obtain the desired result.

Compared to other works such as Bodin & Macris (2021); Adlam & Pennington (2020a), our approach yields smaller sizes of linear-pencils to handle, which in turn yields a smaller set of algebraic equations. One of the ingredient of our method consists in considering a multiple-stage approach where the trace of some random blocks can be calculated in different parts (See the random feature model for example in Appendix D.4). However, the question of finding the simplest linear-pencil remains open and interesting to investigate.

### B.1    LIMITING TRACES OF THE TEST ERROR

LIMITING TRACE FOR $\tilde{f}_1$ AND $f_0$

We construct a linear-pencil $M_1$ as follow (with $Z$ the random matrix into consideration)

$$M_1 = \begin{pmatrix} 0 & 0 & 0 & -yI & 0 & Z^T \\ 0 & 0 & 0 & 0 & Z & I \\ 0 & 0 & 0 & U & I & 0 \\ -xI & 0 & U & -xyV^* & 0 & 0 \\ 0 & Z^T & I & 0 & 0 & 0 \\ Z & I & 0 & 0 & 0 & 0 \end{pmatrix} \tag{96}$$

The inverse of this block-matrix contains the terms in the traces of $\tilde{f}_1$ and $f_0$. To see this, let's calculate the inverse of $M_1$ by splitting it first into other "flattened" blocks:

$$M_1 = \begin{pmatrix} 0 & B_y^T \\ B_x & D \end{pmatrix} \implies M_1^{-1} = \begin{pmatrix} -B_x^{-1} D B_y^{T-1} & B_x^{-1} \\ B_y^{T-1} & 0 \end{pmatrix} \tag{97}$$

Where $B_x$ and $D$ are given by

$$B_x = \begin{pmatrix} -xI & 0 & U \\ 0 & Z^T & I \\ Z & I & 0 \end{pmatrix} \qquad D = \begin{pmatrix} -xyV^* & 0 & 0 \\ 0 & 0 & 0 \\ 0 & 0 & 0 \end{pmatrix} \tag{98}$$

then to calculate the inverse of $B_x$, notice first its lower right-hand sub-block has inverse

$$\begin{pmatrix} Z^T & I \\ I & 0 \end{pmatrix}^{-1} = \begin{pmatrix} 0 & I \\ I & -Z^T \end{pmatrix} \tag{99}$$

Which lead us to the following inverse using the block-matrix inversion formula (the dotted terms aren't required):

$$B_x^{-1} = \begin{pmatrix} G(x) & -G(x)U & G(x)UZ^T \\ -ZG(x) & \ldots & I_n - ZG(x)UZ^T \\ Z^T ZG(x) & \ldots & \ldots \end{pmatrix} \tag{100}$$

With $g_d^{\langle ij \rangle}$ the trace of the squared sub-block $(M_1^{-1})^{\langle ij \rangle}$ divided by the size of the block $(ij)$, we find the desired functions

$$\tilde{f}_1(x, y) = \lim_{d \to +\infty} -g_d^{\langle 11 \rangle} \tag{101}$$

$$f_0(x) = \lim_{d \to +\infty} -g_d^{\langle 15 \rangle} \qquad \text{OR} \qquad f_0(y) = \lim_{d \to +\infty} -g_d^{\langle 51 \rangle} \tag{102}$$

Let's now consider $g$ the limiting value of $g_d$, and calculate the mapping $\eta(g)$:

$$\eta(g) = \begin{pmatrix} 0 & 0 & 0 & 0 & \phi g^{\langle 26 \rangle} & 0 \\ 0 & 0 & 0 & 0 & 0 & g^{\langle 15 \rangle} \\ 0 & 0 & 0 & 0 & 0 & 0 \\ 0 & 0 & 0 & 0 & 0 & 0 \\ \phi g^{\langle 62 \rangle} & 0 & 0 & 0 & \phi g^{\langle 22 \rangle} & 0 \\ 0 & g^{\langle 51 \rangle} & 0 & 0 & 0 & g^{\langle 11 \rangle} \end{pmatrix} \tag{103}$$

So we can calculate the matrix $\Pi(M_1)$ such that the elements of $g$ are the limiting trace of the squared sub-blocks of $(\Pi(M_1))^{-1}$ (divided by the block-size) following the steps of the result in App. E:

$$\Pi(M_1) = \begin{pmatrix} 0 & 0 & 0 & -yI & -\phi g^{\langle 26 \rangle} I & 0 \\ 0 & 0 & 0 & 0 & 0 & (1 - g^{\langle 15 \rangle})I \\ 0 & 0 & 0 & U & I & 0 \\ -xI & 0 & U & -xyV^* & 0 & 0 \\ -\phi g^{\langle 62 \rangle} I & 0 & I & 0 & -\phi g^{\langle 22 \rangle} I & 0 \\ 0 & (1 - g^{\langle 51 \rangle})I & 0 & 0 & 0 & -g^{\langle 11 \rangle} I \end{pmatrix} \tag{104}$$

Therefore, there remains to compute the inverse of $\Pi(M_1)$. We split again $\Pi(M_1)$ as flattened sub-blocks to make the calculation easier

$$\Pi(M_1) = \begin{pmatrix} 0 & \tilde{B}_y^T \\ \tilde{B}_x & \tilde{D} \end{pmatrix} \implies \Pi(M_1)^{-1} = \begin{pmatrix} -\tilde{B}_x^{-1} \tilde{D} (\tilde{B}_y^{-1})^T & \tilde{B}_x^{-1} \\ (\tilde{B}_y^{-1})^T & 0 \end{pmatrix} \tag{105}$$

With the three block-matrices

$$\tilde{B}_x = \begin{pmatrix} -xI & 0 & U \\ -g^{\langle 62 \rangle} \phi I & 0 & I \\ 0 & (1 - g^{\langle 51 \rangle})I & 0 \end{pmatrix} \quad \tilde{B}_y = \begin{pmatrix} -xI & 0 & U \\ -g^{\langle 26 \rangle} \phi I & 0 & I \\ 0 & (1 - g^{\langle 15 \rangle})I & 0 \end{pmatrix} \tag{106}$$

$$\tilde{D} = \begin{pmatrix} -xyV^* & 0 & 0 \\ 0 & -g^{\langle 22 \rangle} \phi I & 0 \\ 0 & 0 & -g^{\langle 11 \rangle} I \end{pmatrix} \tag{107}$$

A straightforward application of the block-matrix inversion formula yields inverse of $\tilde{B}_x$

$$\tilde{B}_x^{-1} = \begin{pmatrix} (\phi g^{\langle 62 \rangle} U - xI)^{-1} & -U(\phi g^{\langle 62 \rangle} U - xI)^{-1} & 0 \\ 0 & 0 & (1 - g^{\langle 51 \rangle})^{-1} I \\ \phi g^{\langle 62 \rangle} (\phi g^{\langle 62 \rangle} U - xI)^{-1} & -x(\phi g^{\langle 62 \rangle} U - xI)^{-1} & 0 \end{pmatrix} \tag{108}$$

Therefore, we retrieve the following close set of equations:

$$g^{\langle 11 \rangle} = \mathrm{Tr}_d \left[ (g^{\langle 62 \rangle} \phi U - xI)^{-1} (xyV^* + g^{\langle 22 \rangle} \phi U^2)(g^{\langle 26 \rangle} \phi U - yI)^{-1} \right] \tag{109}$$

$$g^{\langle 22 \rangle} = g^{\langle 11 \rangle} (1 - g^{\langle 15 \rangle})^{-1} (1 - g^{\langle 51 \rangle})^{-1} \tag{110}$$

$$g^{\langle 26 \rangle} = (1 - g^{\langle 51 \rangle})^{-1} \tag{111}$$

$$g^{\langle 15 \rangle} = -\mathrm{Tr}_d \left[ U(g^{\langle 62 \rangle} \phi U - xI)^{-1} \right] \tag{112}$$

These equations can be simplified slightly by removing $g^{\langle 22 \rangle}, g^{\langle 26 \rangle}$ and introducing $q^{\langle 15 \rangle}$:

$$g^{\langle 11 \rangle} = \mathrm{Tr}_d \left[ (\phi U - xq^{\langle 15 \rangle} I)^{-1} (xyq^{\langle 15 \rangle} q^{\langle 51 \rangle} V^* + g^{\langle 11 \rangle} \phi U^2)(\phi U - yq^{\langle 51 \rangle} I)^{-1} \right] \tag{113}$$

$$q^{\langle 15 \rangle} = \mathrm{Tr}_d \left[ (\phi U - xq^{\langle 15 \rangle} I + q^{\langle 15 \rangle} U)(\phi U - xq^{\langle 15 \rangle} I)^{-1} \right] \tag{114}$$

$$g^{\langle 15 \rangle} = 1 - q^{\langle 15 \rangle} \tag{115}$$

Let $\zeta_x = -xq^{\langle 15 \rangle}$, or by symmetry $\zeta_y = -yq^{\langle 51 \rangle}$, then using the fact that $\tilde{f}_1(x, y) = -g^{\langle 11 \rangle}$ and $f_0(x) = -g^{\langle 15 \rangle}$ we find the system of equations

$$\tilde{f}_1(x, y) = \mathrm{Tr}_d \left[ (\phi U + \zeta_x I)^{-1} (\zeta_x \zeta_y V^* + \tilde{f}_1(x, y) \phi U^2)(\phi U + \zeta_y I)^{-1} \right] \tag{116}$$

$$f_0(x) = - \left( 1 + \frac{\zeta_x}{x} \right) \tag{117}$$

$$\zeta_z = -z + \mathrm{Tr}_d \left[ \zeta_z U(\phi U + \zeta_z I)^{-1} \right] \tag{118}$$

**Remark:** As a byproduct of this analysis, notice the term $g^{\langle 62 \rangle} = (q^{\langle 15 \rangle})^{-1} = \frac{-x}{\zeta_x}$. In fact we have:

$$g^{\langle 62 \rangle} = \text{Tr}_n \left[ I_n - ZG(x)UZ^T \right] \tag{119}$$

$$= 1 - \text{Tr}_n \left[ Z(AA^T Z^T Z - xI)^{-1} AA^T Z^T \right] \tag{120}$$

$$= 1 - \text{Tr}_n \left[ (ZAA^T Z^T - xI)^{-1} ZAA^T Z^T \right] \tag{121}$$

$$= 1 - \text{Tr}_n \left[ (\hat{X}\hat{X}^T - xI)^{-1}(\hat{X}\hat{X}^T - xI_n + xI_n) \right] \tag{122}$$

$$= -x\text{Tr}_n \left[ (\hat{X}\hat{X}^T - xI)^{-1} \right] \tag{123}$$

So if we let $m(x) = \text{Tr}_n \left[ (\hat{X}\hat{X}^T - xI)^{-1} \right]$ the trace of the resolvent of the student data matrix, we find that $m(x) = \zeta_x^{-1}$. This can be useful for analyzing the eigenvalues as in Appendix D.2.1.

LIMITING TRACE FOR $f_2$

As before, we construct a second linear-pencil $M_2$ with $Z$ the random matrix component into consideration

$$M_2 = \begin{pmatrix} I & 0 & 0 & 0 \\ -zV^* & -zI & 0 & U \\ 0 & 0 & Z^T & I \\ 0 & Z & I & 0 \end{pmatrix} \tag{124}$$

The former flattened block $B_z$ can be recognized in the lower right-hand side of $M_2$, thus we can use the block matrix-inversion formula and get:

$$M_2^{-1} = \begin{pmatrix} I & 0 & 0 & 0 \\ zG(z)V^* & & & \\ -zZG(z)V^* & & B_z^{-1} & \\ zZ^T ZG(z)V^* & & & \end{pmatrix} \tag{125}$$

Now it is clear that we can express $f_2(z) = c_0 + \lim_{d \to +\infty} g_d^{\langle 21 \rangle}$. Following the steps of App. E we calculate the mapping

$$\eta(g) = \begin{pmatrix} 0 & 0 & 0 & 0 \\ 0 & 0 & 0 & 0 \\ 0 & \phi g^{\langle 34 \rangle} & 0 & 0 \\ 0 & 0 & g^{\langle 23 \rangle} & 0 \end{pmatrix} \tag{126}$$

Which in returns enable us to calculate $\Pi(M_2)$

$$\Pi(M_2) = \begin{pmatrix} I & 0 & 0 & 0 \\ -zV^* & -zI & 0 & U \\ 0 & -g^{\langle 34 \rangle}\phi I & 0 & I \\ 0 & 0 & (1 - g^{\langle 23 \rangle})I & 0 \end{pmatrix} \tag{127}$$

To compute the inverse of $\Pi(M_2)$, the block-matrix is first split with the sub-block $\tilde{B}_z$ defined as follow

$$\tilde{B}_z = \begin{pmatrix} -zI & 0 & U \\ -g^{\langle 34 \rangle}\phi I & 0 & I \\ 0 & (1 - g^{\langle 23 \rangle})I & 0 \end{pmatrix} \qquad \Pi(M_2) = \begin{pmatrix} I & 0 & 0 & 0 \\ -zV^* & & & \\ 0 & & \tilde{B}_z & \\ 0 & & & \end{pmatrix} \tag{128}$$

A straightforward application of the block-matrix inversion formula yields the inverse of $\tilde{B}_z$:

$$\tilde{B}_z^{-1} = \begin{pmatrix} (g^{\langle 34 \rangle}\phi U - zI)^{-1} & -U(g^{\langle 34 \rangle}\phi U - zI)^{-1} & 0 \\ 0 & 0 & (1 - g^{\langle 23 \rangle})^{-1}I \\ g^{\langle 34 \rangle}\phi(g^{\langle 34 \rangle}\phi U - zI)^{-1} & -z(g^{\langle 34 \rangle}\phi U - zI)^{-1} & 0 \end{pmatrix} \tag{129}$$

Hence we can derive the inverse

$$\Pi(M_2)^{-1} = \begin{pmatrix} I & 0 & 0 & 0 \\ z(g^{\langle 34 \rangle}\phi U - zI)^{-1}V^* & & & \\ 0 & & \tilde{B}_z^{-1} & \\ zg^{\langle 34 \rangle}\phi(g^{\langle 34 \rangle}\phi U - zI)^{-1}V^* & & & \end{pmatrix} \tag{130}$$

Eventually, using the fixed-point result on linear-pencils, we derive the set of equations

$$g^{\langle 21 \rangle} = \text{Tr}_d \left[ zV^*(g^{\langle 34 \rangle} \phi U - zI)^{-1} \right] \tag{131}$$

$$g^{\langle 34 \rangle} = (1 - g^{\langle 23 \rangle})^{-1} \tag{132}$$

$$g^{\langle 23 \rangle} = -\text{Tr}_d \left[ U(g^{\langle 34 \rangle} \phi U - zI)^{-1} \right] \tag{133}$$

$$g^{\langle 41 \rangle} = \text{Tr}_d \left[ zg^{\langle 34 \rangle} \phi (g^{\langle 34 \rangle} \phi U - zI)^{-1} V^* \right] \tag{134}$$

$$g^{\langle 22 \rangle} = \text{Tr}_d \left[ (g^{\langle 34 \rangle} \phi U - zI)^{-1} \right] \tag{135}$$

$$\tag{136}$$

In fact, it is a straightforward to see that $g^{\langle 23 \rangle}, g^{\langle 34 \rangle}$ follows the same equations as the former $g^{\langle 15 \rangle}, g^{\langle 26 \rangle}$ in the previous subsection, therefore $g^{\langle 23 \rangle} = g^{\langle 15 \rangle} = 1 - q^{\langle 15 \rangle} = 1 + \frac{\zeta_z}{z}$, and thus $g^{\langle 34 \rangle} = -\frac{z}{\zeta_z}$ Eventually we get $g^{\langle 21 \rangle} = -\text{Tr}_d \left[ \zeta_z V^*(\phi U + \zeta_z I)^{-1} \right]$ so in the limit $d \to \infty$:

$$f_2(z) = c_0 - \text{Tr}_d \left[ \zeta_z V^*(\phi U + \zeta_z I)^{-1} \right] \tag{137}$$

## B.2 LIMITING TRACES FOR THE TRAINING ERROR

### LIMITING TRACE FOR $h_1$

A careful attention to the linear-pencil $M_1$ shows that the terms in the trace of $\tilde{h}_1$ are actually given by the location $g^{\langle 22 \rangle}$. We have to be careful also of the fact that $(M_1^{-1})^{\langle 22 \rangle}$ is a block matrix of size $n \times n$, so it is already divided by the size $n$ (and not $d$). Hence we simply have with $\eta_z = \frac{-z}{\zeta_z}$:

$$\tilde{h}_1(x, y) = -g^{\langle 22 \rangle} = \frac{-x}{\zeta_x} \frac{-y}{\zeta_y} f_1(x, y) = \eta_x \eta_y f_1(x, y) \tag{138}$$

### LIMITING TRACE FOR $h_2$

In the case of $h_2$, we need the specific term provided by the linear-pencil $M_2$ by the location $g^{\langle 41 \rangle}$ with $\phi h_2(z) = c_0 \phi + g^{\langle 41 \rangle}$

For $h_2$ we use the linear pencil for $f_2$, but instead of using $g^{\langle 21 \rangle}$ we use $h_2 = c_0 \phi + g^{\langle 41 \rangle}$. We find:

$$g^{\langle 41 \rangle} = z\phi \text{Tr}_d \left[ V^*(\phi U + \zeta_z I)^{-1} \right] \tag{139}$$

$$= \phi \frac{z}{\zeta_z} \text{Tr}_d \left[ \zeta_z V^*(\phi U + \zeta_z I)^{-1} \right] \tag{140}$$

$$= \phi \frac{z}{\zeta_z}(c_0 - f_2(z)) \tag{141}$$

Hence:

$$h_2(z) = c_0 \left( 1 - \frac{-z}{\zeta_z} \right) + \frac{-z}{\zeta_z} f_2(z) = \eta_z(c_0 f_0(z) + f_2(z)) \tag{142}$$

### LIMITING TRACE FOR $h_0$

Finally for $h_0$ we use again the linear pencil $M_2$ with:

$$\text{Tr}_d \left[ zG(z) \right] = zg^{\langle 22 \rangle} = -\text{Tr}_d \left[ \zeta_z(\phi U + \zeta_z I)^{-1} \right] \tag{143}$$

$$= -\text{Tr}_d \left[ (\zeta_z + \phi U - \phi U)(\phi U + \zeta_z I)^{-1} \right] \tag{144}$$

$$= -1 + \phi \text{Tr}_d \left[ U(\phi U + \zeta_z I)^{-1} \right] \tag{145}$$

$$= -1 + \frac{\phi}{\zeta_z} \text{Tr}_d \left[ \zeta_z U(\phi U + \zeta_z I)^{-1} \right] \tag{146}$$

$$= -1 + \frac{\phi}{\zeta_z}(\zeta_z + z) \tag{147}$$

Therefore:

$$h_0(z) = \left( 1 - \frac{-z}{\zeta_z} \right) = - \left( 1 + \frac{\zeta_z}{z} \right) \frac{-z}{\zeta_z} = \eta_z f_0(z) \tag{148}$$

## C   Other limiting expressions

In this section we bring the sketch of proofs of additional expressions seen in the main results.

### C.1   Expression with dual counterpart matrices $U_\star$ and $V_\star$

The former functionals $f_2$ and $\tilde{f}_1$ can be rewritten as:

$$f_2(z) = c_0 - \text{Tr}_d \left[ \zeta_z V^* (\phi U + \zeta_z I)^{-1} \right] \tag{149}$$

$$= c_0 - \text{Tr}_d \left[ (\zeta_z I + \phi U - \phi U) V^* (\phi U + \zeta_z I)^{-1} \right] \tag{150}$$

$$= c_0 - \text{Tr}_d \left[ V^* \right] + \text{Tr}_d \left[ \phi A^T V^* (\phi U + \zeta_z I)^{-1} A^T \right] \tag{151}$$

$$= c_0 - c_0 + \text{Tr}_d \left[ \phi A^T B \beta^* \beta^{*T} B^T A (U_\star + \zeta_z I)^{-1} \right] \tag{152}$$

$$= \text{Tr}_n \left[ (\Xi \beta^* \beta^{*T} \Xi^T)(U_\star + \zeta_z I)^{-1} \right] \tag{153}$$

With similar steps using:

$$\zeta_x V^* \zeta_y = -(\zeta_x I + \phi U) V^* (\zeta_y I + \phi U) + \zeta_x V^* (\zeta_y I + \phi U) + (\zeta_x I + \phi U) V^* \zeta_y + \phi^2 U V^* U \tag{154}$$

We find:

$$\tilde{f}_1(x, y) = -c_0 + \text{Tr}_d \left[ \zeta_x V^* (\zeta_y I + \phi U)^{-1} \right] + \text{Tr}_d \left[ (\zeta_x I + \phi U)^{-1} V^* \zeta_y \right] \tag{155}$$

$$+ \text{Tr}_d \left[ (\phi U + \zeta_x I)^{-1} (\phi^2 U V^* U + \tilde{f}_1(x, y) \phi U^2)(\phi U + \zeta_y I)^{-1} \right] \tag{156}$$

$$= c_0 - f_2(x) - f_2(y) \tag{157}$$

$$+ \text{Tr}_n \left[ (U_\star + \zeta_x I)^{-1} ((\Xi \beta^* \beta^{*T} \Xi^T) + \tilde{f}_1(x, y) U_\star) U_\star (U_\star + \zeta_y I)^{-1} \right] \tag{158}$$

Hence in fact:

$$f_1(x, y) = \text{Tr}_n \left[ (U_\star + \zeta_x I)^{-1} ((\Xi \beta^* \beta^{*T} \Xi^T) + \tilde{f}_1(x, y) U_\star) U_\star (U_\star + \zeta_y I)^{-1} \right] \tag{159}$$

Finally, we have using the push-through identity and the cyclicity of the trace:

$$\zeta_z = -z + \text{Tr}_d \left[ \zeta_z A A^T (\phi A A^T + \zeta_z I)^{-1} \right] \tag{160}$$

$$= -z + \text{Tr}_d \left[ \zeta_z A (\phi A^T A + \zeta_z I)^{-1} A^T \right] \tag{161}$$

$$= -z + \text{Tr}_n \left[ \zeta_z U_\star (U_\star + \zeta_z I)^{-1} \right] \tag{162}$$

### C.2   Limiting trace of $Z^T Z V^*$

Here we show another way in which our random matrix result can be used to infer the result on the limiting trace $\text{Tr}_d \left[ Z^T Z V^* \right]$. To this end, we can design the linear-pencil:

$$M_3 = \begin{pmatrix} I & -V^* & 0 & 0 \\ 0 & I & Z^T & 0 \\ 0 & 0 & I & Z \\ 0 & 0 & 0 & I \end{pmatrix} \tag{163}$$

It is straightforward to calculate the inverse of the sub-matrix:

$$\begin{pmatrix} I & Z^T & 0 \\ 0 & I & Z \\ 0 & 0 & I \end{pmatrix}^{-1} = \begin{pmatrix} I & -Z^T & Z^T Z \\ 0 & I & -Z \\ 0 & 0 & I \end{pmatrix} \tag{164}$$

So that:

$$M_3^{-1} = \begin{pmatrix} I & V^* & -Z^T V^* & V^* Z^T Z \\ 0 & I & -Z^T & Z^T Z \\ 0 & 0 & I & -Z \\ 0 & 0 & 0 & I \end{pmatrix} \tag{165}$$

At this point, it is clear that the quantity of interest is provided by the term $g^{\langle 14 \rangle}$ of the linear-pencil $M_3$. We find calculate further:

$$\eta(g) = \begin{pmatrix} 0 & 0 & 0 & 0 \\ 0 & 0 & 0 & \phi g^{\langle 33 \rangle} \\ 0 & 0 & g^{\langle 42 \rangle} & 0 \\ 0 & 0 & 0 & 0 \end{pmatrix} \tag{166}$$

Based on the inverse of $M_3$, we can already predict that $g^{\langle 33 \rangle} = 1$ and $g^{\langle 42 \rangle} = 0$. Hence:

$$\Pi(M_3) = \begin{pmatrix} I & -V^* & 0 & 0 \\ 0 & I & 0 & -\phi I \\ 0 & 0 & I & 0 \\ 0 & 0 & 0 & I \end{pmatrix} \implies \Pi(M_3)^{-1} = \begin{pmatrix} I & V^* & 0 & \phi V^* \\ 0 & I & 0 & \phi I \\ 0 & 0 & I & 0 \\ 0 & 0 & 0 & I \end{pmatrix} \tag{167}$$

Finally we obtain $g^{\langle 14 \rangle} = \mathrm{Tr}_d\left[\phi V^*\right]$, and hence $\mathrm{Tr}_d\left[Z^T Z V^*\right] = \phi \mathrm{Tr}_d\left[V^*\right]$.

# D  APPLICATIONS AND CALCULATION DETAILS

## D.1  MISMATCHED RIDGELESS REGRESSION OF A NOISY LINEAR FUNCTION

**Target function** Here we consider a slightly more complicated version of the former example where we let $y(x_0, x_1) = r\left(x_0^T \beta_0^* + x_1^T \beta_1^*\right) + \sigma \epsilon$ and still averaged over $\beta_0 \sim \mathcal{N}(0, I_{\gamma p})$ and $\beta_1 \sim \mathcal{N}(0, I_{(1-\gamma)p})$ with $x_0 \in \mathbb{R}^{\gamma p}, x_1 \in \mathbb{R}^{(1-\gamma)p}$. We let again $d = p + q$ and $\psi = \frac{p}{d}$ and $\phi_0 = \frac{p}{q}$. Therefore the former relation still holds $\phi = \frac{n}{d} = \frac{n}{p}\frac{p}{d} = \phi_0 \psi$. Similarly, we derive a block-matrix $B$ and compute $V^*$:

$$B = \begin{pmatrix} r\sqrt{\frac{d}{p}} I_{\gamma p} & 0 & 0 \\ 0 & r\sqrt{\frac{d}{p}} I_{(1-\gamma)p} & 0 \\ 0 & 0 & \sigma\sqrt{\frac{d}{q}} I_q \end{pmatrix} \implies V^* = \begin{pmatrix} \frac{r^2}{\psi} I_{\gamma p} & 0 & 0 \\ 0 & \frac{r^2}{\psi} I_{(1-\gamma)q} & 0 \\ 0 & 0 & \frac{\sigma^2}{1-\psi} I_q \end{pmatrix} \tag{168}$$

So that with the splitting $Z = \left(\sqrt{\frac{p}{d}} X_0 | \sqrt{\frac{p}{d}} X_1 | \sqrt{\frac{q}{d}} \Sigma\right)$, and $\beta^{*T} = \left(\beta_0^{*T} | \beta_1^{*T} | \beta_2^{*T}\right)$, and with $\xi = \Sigma \beta_2^*$:

$$Y = ZB\beta^* = r(X_0 \beta_0^* + X_1 \beta_1^*) + \sigma \xi \tag{169}$$

**Estimator** Following the same steps, we construct $A$ and $U$ with

$$A = \begin{pmatrix} \sqrt{\frac{d}{\gamma p}} I_{\gamma p} \\ 0_{(1-\gamma)p \times \gamma d} \\ 0_{q \times \gamma d} \end{pmatrix} \implies U = \begin{pmatrix} \frac{1}{\gamma \psi} I_{\gamma p} & 0 & 0 \\ 0 & 0 & 0 \\ 0 & 0 & 0 \end{pmatrix} \tag{170}$$

So that we get the linear estimator $\hat{Y}_t$

$$\hat{Y}_t = ZA\beta_t = \frac{1}{\sqrt{\gamma}} X_0 \beta_t \tag{171}$$

**Analytic result** as $U$ and $V^*$ commute again, the joint probability distribution can be derived:

$$\mathcal{P}\left(u = \frac{1}{\gamma \psi}, v = \frac{r^2}{\psi}\right) = \gamma \psi \tag{172}$$

$$\mathcal{P}\left(u = 0, v = \frac{r^2}{\psi}\right) = (1 - \gamma)\psi \tag{173}$$

$$\mathcal{P}\left(u = 0, v = \frac{\sigma^2}{(1 - \psi)}\right) = 1 - \psi \tag{174}$$

Therefore, in the regime $\lambda = 0$, with $\kappa = \frac{\phi_0}{\gamma}$, a calculation leads to the following result (dubbed the "mismatched model" in Hastie et al. (2019))

$$\mathcal{E}_{\text{gen}}(+\infty) = \tilde{f}_1 = \begin{cases} \frac{\kappa}{\kappa-1}(\sigma^2 + (1-\gamma)r^2) & (\kappa > 1) \\ \frac{1}{1-\kappa}\sigma^2 + r^2\gamma(1-\kappa) & (\kappa < 1) \end{cases} \tag{175}$$

### D.2   NON ISOTROPIC MODEL

We have the joint probabilities $P(u = \alpha^{-i}, v = 1) = \frac{1}{p} = \gamma$ for $i \in \{0, \ldots, p-1\}$ and $\lambda = 0$. Then:

$$\tilde{f}_1 = \frac{1}{p} \sum_{i=0}^{p-1} \frac{\tilde{f}_1 \phi + (\alpha^i \zeta)^2}{(\phi + \alpha^i \zeta)^2} \tag{176}$$

$$\zeta = \frac{1}{p} \sum_{i=0}^{p-1} \frac{\zeta}{\phi + \alpha^i \zeta} \tag{177}$$

$$f_2 = c_0 - \frac{1}{p} \sum_{i=0}^{p-1} \frac{\zeta \alpha^i}{\phi + \alpha^i \zeta} \tag{178}$$

So either $\zeta = 0$ and thus $\tilde{f}_1 = 0$, or $\zeta \neq 0$ and:

$$\tilde{f}_1 = \left( 1 - \frac{1}{p} \sum_{i=0}^{p-1} \frac{\phi}{(\phi + \alpha^i \zeta)^2} \right)^{-1} \frac{1}{p} \sum_{i=0}^{p-1} \frac{(\alpha^i \zeta)^2}{(\phi + \alpha^i \zeta)^2} \tag{179}$$

$$1 = \frac{1}{p} \sum_{i=0}^{p-1} \frac{1}{\phi + \alpha^i \zeta} \tag{180}$$

Writing further down $(\alpha^i \zeta)^2 = (\alpha^i \zeta + \phi - \phi)^2 = (\alpha^i \zeta + \phi)^2 - 2\phi(\alpha^i \zeta + \phi) + \phi^2$ we get:

$$\frac{1}{p} \sum_{i=0}^{p-1} \frac{(\alpha^i \zeta)^2}{(\phi + \alpha^i \zeta)^2} = 1 - 2\phi \frac{1}{p} \sum_{i=0}^{p-1} \frac{1}{\phi + \alpha^i \zeta} + \phi^2 \frac{1}{p} \sum_{i=0}^{p-1} \frac{1}{(\phi + \alpha^i \zeta)^2} \tag{181}$$

$$= 1 - 2\phi + \phi^2 \frac{1}{p} \sum_{i=0}^{p-1} \frac{1}{(\phi + \alpha^i \zeta)^2} \tag{182}$$

$$= (1 - \phi) - \phi \left( 1 - \frac{1}{p} \sum_{i=0}^{p-1} \frac{\phi}{(\phi + \alpha^i \zeta)^2} \right) \tag{183}$$

So:

$$\tilde{f}_1 = (1 - \phi) \left( 1 - \frac{1}{p} \sum_{i=0}^{p-1} \frac{\phi}{(\phi + \alpha^i \zeta)^2} \right)^{-1} - \phi \tag{184}$$

Now injecting the expression for $\zeta$:

$$1 - \frac{1}{p} \sum_{i=0}^{p-1} \frac{\phi}{(\phi + \alpha^i \zeta)^2} = \frac{1}{p} \sum_{i=0}^{p-1} \left( \frac{1}{\phi + \alpha^i \zeta} - \frac{\phi}{(\phi + \alpha^i \zeta)^2} \right) \tag{185}$$

$$= \frac{1}{p} \sum_{i=0}^{p-1} \frac{\alpha^i \zeta}{(\phi + \alpha^i \zeta)^2} \tag{186}$$

Hence the formula

$$\mathcal{E}_{\text{gen}}(\infty) = (1 - \phi) \left( \frac{1}{p} \sum_{i=0}^{p-1} \frac{\alpha^i \zeta}{(\phi + \alpha^i \zeta)^2} \right)^{-1} - \phi \tag{187}$$

**Asymptotic limit:** Let's consider the behavior of the generalisation error when $\alpha \to \infty$. Let's consider the potential solution for some $k \in \{0, \ldots, p-1\}$:

$$\zeta^k = \frac{c_k}{\alpha^k}(1 + o_\alpha(1)) \tag{188}$$

for some constant $c_k$. Then:

$$p = \sum_{i=0}^{p-1} \frac{1}{\phi + c_k \alpha^{i-k}(1 + o_\alpha(1))} = \frac{1}{\phi + c_k} + \frac{k}{\phi} + o_\alpha(1) \tag{189}$$

Hence we choose:

$$c_k = \phi\left(\frac{1}{p\phi - k} - 1\right) \tag{190}$$

Because $\mathcal{E}_{\text{gen}}(\infty) \geq 0$, we need to enforce $\zeta^k > 0$ which leads to the condition $\frac{1}{p\phi - k} - 1 \geq 0$, that is $1 \geq p\phi - k > 0$. So in fact it implies $\phi \in \left]\frac{k}{p}, \frac{k+1}{p}\right]$, so $\zeta^k$ can only be a solution for $\phi$ in this range. Therefore we can consider the solution $\zeta(\phi) = \sum_{i=0}^{p-1} \mathbb{1}_{]\frac{k}{p}; \frac{k+1}{p}[}(\phi)\zeta^k(\phi)$. Then notice:

$$\sum_{i=0}^{p-1} \frac{\alpha^i \zeta^k}{(\phi + \alpha^i \zeta^k)^2} = \frac{c_k}{(c_k + \phi)^2} + o_\alpha(1) = -p^2\left(\phi - \frac{k}{p}\right)\left(\phi - \frac{k+1}{p}\right) + o_\alpha(1) \tag{191}$$

and thus for $\phi \in [0,1] \setminus \frac{k}{p}\mathbb{Z}$:

$$\mathcal{E}_{\text{gen}}(\infty) = \sum_{k=0}^{p-1} \frac{\phi(1-\phi)}{p\left(\phi - \frac{k}{p}\right)\left(\frac{k+1}{p} - \phi\right)}\mathbb{1}_{]\frac{k}{p}; \frac{k+1}{p}[}(\phi) - \phi + o_\alpha(1) \tag{192}$$

So we clearly see that in the limit of $\alpha$ large, the test error approaches a function with two roots at the denominator.

**Evolution:**

$$\tilde{f}_1(x,y) = \frac{1}{p}\sum_{i=0}^{p-1} \frac{\tilde{f}_1(x,y)\phi + \alpha^{2i}\zeta_x\zeta_y}{(\phi + \alpha^i\zeta_x)(\phi + \alpha^i\zeta_y)} \tag{193}$$

$$\zeta_z = -z + \frac{1}{p}\sum_{i=0}^{p-1} \frac{\zeta_z}{\phi + \alpha^i\zeta_z} \tag{194}$$

$$f_2(z) = c_0 - \frac{1}{p}\sum_{i=0}^{p-1} \frac{\alpha^i\zeta_z}{\phi + \alpha^i\zeta_z} \tag{195}$$

In particular $f_2$ is given by:

$$f_2(z) = c_0 - 1 + \frac{\phi}{p}\sum_{i=0}^{p-1} \frac{1}{\phi + \alpha^i\zeta_z} = c_0 - 1 + \phi\zeta_z\left(1 + \frac{z}{\zeta_z}\right) \tag{196}$$

and $\tilde{f}_1$ is given by:

$$\tilde{f}_1(x,y) = \frac{\frac{1}{p}\sum_{i=0}^{p-1} \frac{\alpha^{2i}\zeta_x\zeta_y}{(\phi + \alpha^i\zeta_x)(\phi + \alpha^i\zeta_y)}}{1 - \frac{\phi}{p}\sum_{i=0}^{p-1} \frac{1}{(\phi + \alpha^i\zeta_x)(\phi + \alpha^i\zeta_y)}} \tag{197}$$

### D.2.1 Eigenvalue distribution

In our figures, we look at the log-eigenvalue distribution of the student data $\rho_{\log \lambda}$ as it provides the most natural distributions on a log-scale basis. So in fact, if we plot the curve $y(x) = \rho_{\log \lambda}(x)$ we have:

$$y(x) = \rho_{\log \lambda}(x) = \frac{\partial}{\partial x}\mathcal{P}(\log \lambda \leq x) \tag{198}$$

$$= \frac{\partial}{\partial x}\mathcal{P}(\lambda \leq e^x) \tag{199}$$

$$= e^x \rho_\lambda(e^x) \tag{200}$$

So in a log-scale basis we have $\rho_{\log \lambda}(\log x) = x\rho_\lambda(x)$. It is interesting to notice the connection with $\eta_x$ for running computer simulations:

$$\rho_{\log \lambda}(\log x) = \frac{x}{\pi}\lim_{\epsilon \to 0^+} m(x + i\epsilon) = \frac{1}{\pi}\lim_{\epsilon \to 0^+} \frac{x + i\epsilon}{\zeta(x + i\epsilon)} = -\frac{1}{\pi}\lim_{\epsilon \to 0^+} \eta_{x+i\epsilon} \tag{201}$$

It is work mentioning that the bulks are further "detached" as $\alpha$ grows as it can be seen in figure 6. Furthermore, bigger $\alpha$ makes the spike more distringuisable.

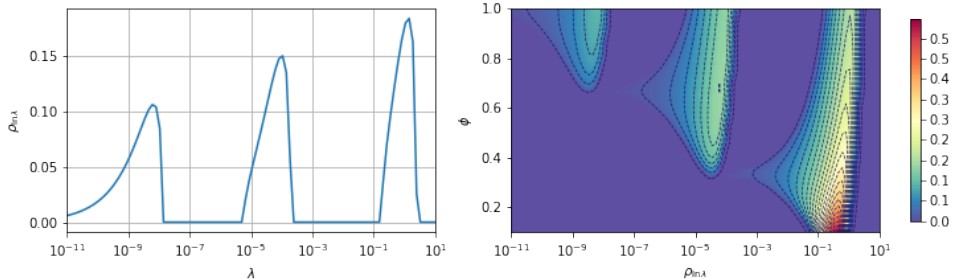

Figure 6: Theoretical (log-)eigenvalue distribution in the non-isotropic ridgeless regression model with $p = 3, \lambda = 10^{-5}, \alpha = 10^4$ with $\phi = 1$ on the left and a range $\phi \in (0, 1)$ on the right heatmap.

### D.3 KERNEL METHODS

Kernel methods are equivalent to solving the following linear regression problem:

$$\beta = \arg\min_{\beta} \sum_{i=1}^{n} \left( \theta_0^T \phi(x_i) - \beta^T \phi(x_i) \right)^2 + \lambda \|\beta\|^2 \tag{202}$$

Where $\phi(x) = (\phi_i(x))_{i \in \mathbb{N}} = (\sqrt{\omega_i} e_i(x))$ for some orthogonal basis $(e_i)_{i \in \mathbb{N}}$. In fact we can consider:

$$A = B = \begin{pmatrix} \sqrt{\omega_1} & 0 & \cdots & 0 \\ 0 & \sqrt{\omega_2} & \cdots & 0 \\ \vdots & \vdots & \ddots & \vdots \\ 0 & 0 & \cdots & \sqrt{\omega_d} \end{pmatrix} \tag{203}$$

and $z_i = (e_1(x_i), \ldots, e_d(x_i))$. Then let's consider the following linear regression problem:

$$\hat{\beta} = \arg\min_{\beta} \|Z (B\beta^* - A\beta)\|^2 + \lambda \|\beta\|^2 \tag{204}$$

$$\mathcal{E}_{\text{gen}}(\hat{\beta}) = \mathbb{E}_z \left[ \left( z^T \left( B\beta^* - A\hat{\beta} \right) \right)^2 \right] \tag{205}$$

This problem is identical to the kernel methods in the situation with a specific $\beta^{*T} = (\theta_{01}, \ldots, \theta_{0d})$. Although $V^*$ and $U$ don't commute with each other, Notice that with $x = y = -\lambda$, due to the diagonal structure of $U$:

$$\tilde{f}_1 = \text{Tr}_d \left[ (\phi U + \zeta I)^{-1} (\zeta^2 V^* + \tilde{f}_1 \phi U^2)(\phi U + \zeta I)^{-1} \right] \tag{206}$$

$$= \frac{1}{d} \sum_{i=1}^{d} [(\zeta^2 V^* + \tilde{f}_1 \phi U^2)(\phi U + \zeta I)^{-2}]_{ii} \tag{207}$$

$$= \frac{1}{d} \sum_{i=1}^{d} (\zeta^2 [V^*]_{ii} + \tilde{f}_1 \phi [U^2]_{ii})(\phi [U]_{ii} + \zeta)^{-2} \tag{208}$$

So in fact we find the self-consistent set of equation with $\mathcal{E}_{\text{gen}}(+\infty) = \tilde{f}_1$:

$$\zeta = \lambda + \frac{1}{d} \sum_{i=1}^{d} \frac{\zeta \omega_i}{\phi \omega_i + \zeta} \tag{209}$$

$$\tilde{f}_1 = \frac{1}{d} \sum_{i=1}^{d} \frac{\tilde{f}_1 \phi \omega_i^2 + \zeta^2 \theta_{0i}^2 \omega_i}{(\phi \omega_i + \zeta)^2} \tag{210}$$

This is precisely the results from equation (78) in Loureiro et al. (2021) (see also Bordelon et al. (2020)) with the change of variables $\lambda(1 + V) \to \zeta$ and $\rho + q - 2m \to \tilde{f}_1$.

### D.4 RANDOM FEATURES EXAMPLE

We get the following matrices $U, V$ with $\tilde{\mu}^2 = \frac{\mu^2}{\psi}, \tilde{\nu}^2 = \frac{\nu^2}{\psi}, \tilde{r}^2 = \frac{r^2}{\psi}, \tilde{\sigma}^2 = \frac{\sigma^2}{1-(1+\psi_0)\psi}$:

$$U = \begin{pmatrix} \tilde{\mu}^2 W W^T & \tilde{\mu}\tilde{\nu}W & 0 \\ \tilde{\mu}\tilde{\nu}W^T & \tilde{\nu}^2 I_N & 0 \\ 0 & 0 & 0 \end{pmatrix} \qquad V = \begin{pmatrix} \tilde{r}^2 I_p & 0 & 0 \\ 0 & 0 & 0 \\ 0 & 0 & \tilde{\sigma}^2 I_q \end{pmatrix} \tag{211}$$

In fact, the matrices $U$ and $V$ do not commute with each other, so we have more involved calculations. First we consider the subspace $F = \mathrm{Ker}(V - \tilde{\sigma}^2 I_q)^{\perp}$. Let's define the matrices:

$$U_F = \begin{pmatrix} \tilde{\mu}^2 W W^T & \tilde{\mu}\tilde{\nu}W \\ \tilde{\mu}\tilde{\nu}W^T & \tilde{\nu}^2 I_N \end{pmatrix} \qquad V_F = \begin{pmatrix} \tilde{r}^2 I_p & 0 \\ 0 & 0 \end{pmatrix} \tag{212}$$

$$U_{F^{\perp}} = (0) \qquad V_{F^{\perp}} = (\tilde{\sigma}^2 I_q) \tag{213}$$

Then, although $U$ and $V$ can't be diagonalized in the same basis, they are still both block-diagonal matrices in the same direct-sum space $\mathbb{R}^d = F \oplus F^{\perp}$, so in fact the following split between the two subspaces $F$ and $F^{\perp}$ holds:

$$\tilde{f}_1 = \mathrm{Tr}_d \left[ (\phi U_F + \zeta_x I)^{-1} \zeta_x \zeta_y V_F (\phi U_F + \zeta_y I)^{-1} \right] \tag{214}$$

$$+ \mathrm{Tr}_d \left[ (\phi U_{F^{\perp}} + \zeta_x I)^{-1} \zeta_x \zeta_y V_{F^{\perp}} (\phi U_{F^{\perp}} + \zeta_y I)^{-1} \right] \tag{215}$$

$$+ \mathrm{Tr}_d \left[ (\phi U + \zeta_x I)^{-1} \tilde{f}_1 \phi U^2 (\phi U + \zeta_y I)^{-1} \right] \tag{216}$$

Now let's define $\kappa_1, \kappa_2, \kappa_3$ such that:

$$\tilde{f}_1 = r^2 \kappa_1 + \tilde{f}_1 (1 - \kappa_2^{-1}) + \sigma^2 \kappa_3 \tag{217}$$

That is to say, we get directly $\tilde{f}_1 = (r^2 \kappa_1 + \sigma^2 \kappa_3)\kappa_2$ and by definition:

$$r^2 \kappa_1 = \mathrm{Tr}_d \left[ (\phi U_F + \zeta_x I)^{-1} \zeta_x \zeta_y V_F (\phi U_F + \zeta_y I)^{-1} \right] \tag{218}$$

$$1 - \frac{1}{\kappa_2} = \mathrm{Tr}_d \left[ (\phi U + \zeta_x I)^{-1} \phi U^2 (\phi U + \zeta_y I)^{-1} \right] \tag{219}$$

$$\sigma^2 \kappa_3 = \mathrm{Tr}_d \left[ (\phi U_{F^{\perp}} + \zeta_x I_q)^{-1} \zeta_x \zeta_y V_{F^{\perp}} (\phi U_{F^{\perp}} + \zeta_y I_q)^{-1} \right] = \sigma^2 \tag{220}$$

So we already know that $\kappa_3 = 1$. Let's focus on $\kappa_1$, we can deal with a linear pencil $M$ such that we would get the desired term. First we define similarly $A_F^T$, the restriction of $A^T$ on the subspace $F$:

$$A_F = \begin{pmatrix} \tilde{\mu}W \\ \tilde{\nu}I_N \end{pmatrix} \implies U_F = A_F A_F^T \tag{221}$$

Then, following the structure of $M_1$ we can construct the following linear-pencil $M$:

$$M = \begin{pmatrix} 0 & 0 & \zeta_y I & A_F \\ 0 & 0 & A_F^T & -\frac{1}{\phi}I \\ \zeta_x I & A_F & -\zeta_x \zeta_y V_F & 0 \\ A_F^T & -\frac{1}{\phi}I & 0 & 0 \end{pmatrix} = \left( \begin{array}{c|c} 0 & B_y \\ \hline B_x & \begin{pmatrix} -\zeta_x \zeta_y V_F & 0 \\ 0 & 0 \end{pmatrix} \end{array} \right) \tag{222}$$

So that:

$$M^{-1} = \left( \begin{array}{c|c} B_x^{-1} \begin{pmatrix} -\zeta_x \zeta_y V_F & 0 \\ 0 & 0 \end{pmatrix} B_y^{-1} & B_x^{-1} \\ \hline B_y^{-1} & 0 \end{array} \right) \tag{223}$$

where:

$$B_x^{-1} = \begin{pmatrix} (\phi U_F + \zeta_x I)^{-1} & \phi(\phi U_F + \zeta_x I)^{-1} A_F \\ A_F^T \phi(\phi U_F + \zeta_x I)^{-1} & (-\frac{1}{\phi}I - \frac{1}{\zeta_y}A_F^T A_F)^{-1} \end{pmatrix} \tag{224}$$

In the above matrices, the sub-blocks $A_F$ and $V_F$ are implicitly flattened, so in fact $M$ is given completely by:

$$M = \begin{pmatrix} 0 & 0 & 0 & \zeta_y I & 0 & \tilde{\mu}W \\ 0 & 0 & 0 & 0 & \zeta_y I & \tilde{\nu}I \\ 0 & 0 & 0 & \tilde{\mu}W^T & \tilde{\nu}I & -\frac{1}{\phi}I \\ \zeta_x I & 0 & \tilde{\mu}W & -\tilde{r}^2 \zeta_x \zeta_y I_p & 0 & 0 \\ 0 & \zeta_x I & \tilde{\nu}I & 0 & 0 & 0 \\ \tilde{\mu}W^T & \tilde{\nu}I & -\frac{1}{\phi}I & 0 & 0 & 0 \end{pmatrix} \tag{225}$$

and therefore, one has to pay attention on the quantity of interest which is given by a sum of two terms:

$$r^2 \kappa_1 = \lim_{d \to +\infty} \left( \frac{p}{d} g^{\langle 11 \rangle} + \frac{N}{d} g^{\langle 22 \rangle} \right) = \psi(g^{\langle 11 \rangle} + \psi_0 g^{\langle 22 \rangle}) \tag{226}$$

Using a Computer-Algebra-System, we get the equations with $\gamma_x, \gamma_y, \delta_x, \delta_y$ defined such that $g^{\langle 36 \rangle} = -\psi \gamma_x \zeta_x$, $g^{\langle 63 \rangle} = -\psi \gamma_y \zeta_y$, $\delta_x = \zeta_x g^{\langle 14 \rangle}$, $\delta_y = \zeta_y g^{\langle 41 \rangle}$:

$$\psi g^{\langle 11 \rangle} = (\zeta_x \zeta_y)^{-1} (\delta_x \delta_y)(r^2 \zeta_x \zeta_y + \mu^2 \psi_0 g^{\langle 33 \rangle}) \tag{227}$$

$$\psi g^{\langle 22 \rangle} = \phi^{-2} (\gamma_x \gamma_y)(\psi g^{\langle 11 \rangle} \mu^2 \nu^2 \phi^2) \tag{228}$$

$$g^{\langle 33 \rangle} = (\zeta_x \zeta_y)(\gamma_x \gamma_y)(\psi g^{\langle 11 \rangle} \mu^2) \tag{229}$$

$$\delta_y = (1 + \gamma_y \mu^2 \psi_0)^{-1} \tag{230}$$

$$\gamma_y = (\mu^2 \delta_y + \phi_0^{-1} \zeta_y + \nu^2)^{-1} \tag{231}$$

So:

$$(1 - \mu^4 \psi_0 (\delta_x \delta_y)(\gamma_x \gamma_y))\psi g^{\langle 11 \rangle} = (\delta_x \delta_y)(r^2) \tag{232}$$

and:

$$\psi g^{\langle 11 \rangle} + \psi_0 \psi g^{\langle 22 \rangle} = \left( 1 + \psi_0 \mu^2 \nu^2 (\gamma_x \gamma_y) \right) \left( \psi g^{\langle 11 \rangle} \right) \tag{233}$$

Hence the result:

$$\kappa_1 = \frac{1 + \nu^2 \mu^2 \psi_0 (\gamma_x \gamma_y)}{1 - \mu^4 \psi_0 (\delta_x \delta_y)(\gamma_x \gamma_y)} (\delta_x \delta_y) \tag{234}$$

Also there remain to use the last equation regarding $\zeta_x$ using the fact that:

$$\zeta_y + y = \mathrm{Tr}_d \left[ \zeta_x U (\phi U + \zeta_x I)^{-1} \right] \tag{235}$$

Notice that we have

$$g^{\langle 63 \rangle} = -\gamma_y \psi \zeta_y = \mathrm{Tr}_N \left[ \left( -\frac{1}{\phi} I - \frac{1}{\zeta_y} A_F^T A_F \right)^{-1} \right] \tag{236}$$

So because $A_F^T A_F = A^T A$:

$$\zeta_y \gamma_y = \phi_0 \zeta_y \mathrm{Tr}_N \left[ (\phi A^T A + \zeta_y I)^{-1} \right] \tag{237}$$

$$= \phi_0 \mathrm{Tr}_N \left[ (\phi A^T A + \zeta_y I - \phi A^T A)(\phi A^T A + \zeta_y I)^{-1} \right] \tag{238}$$

$$= \phi_0 \mathrm{Tr}_N \left[ I - \phi A^T A (\phi A^T A + \zeta_y I)^{-1} \right] \tag{239}$$

$$= \phi_0 \left( 1 - \mathrm{Tr}_N \left[ \phi (\phi U + \zeta_y I)^{-1} U \right] \right) \tag{240}$$

$$= \phi_0 \left( 1 - \frac{\phi_0}{\psi_0 \zeta_y} \mathrm{Tr}_d \left[ \zeta_y U (\phi U + \zeta_y I)^{-1} \right] \right) \tag{241}$$

$$= \phi_0 \left( 1 - \frac{\phi_0}{\psi_0 \zeta_y} (\zeta_y + y) \right) \tag{242}$$

Therefore:

$$\frac{\gamma_y}{\phi_0} \zeta_y = 1 - \frac{\phi_0}{\psi_0} \left( 1 + \frac{y}{\zeta_y} \right) \tag{243}$$

For $\kappa_2$ we can calculate the following expression - which in fact is general and doesn't depend on the specific design of $U$:

$$1 - \frac{1}{\kappa_2} = \mathrm{Tr}_d\left[(\phi U + \zeta_x I)^{-1}\phi U^2(\phi U + \zeta_y I)^{-1}\right] \tag{244}$$

$$= \mathrm{Tr}_d\left[(\phi U + \zeta_x I)^{-1}(\phi U + \zeta_x I - \zeta_x I)U(\phi U + \zeta_y I)^{-1}\right] \tag{245}$$

$$= \mathrm{Tr}_d\left[(I - \zeta_x(\phi U + \zeta_x I)^{-1})U(\phi U + \zeta_y I)^{-1}\right] \tag{246}$$

$$= \mathrm{Tr}_d\left[U(\phi U + \zeta_y I)^{-1} - \zeta_x(\phi U + \zeta_x I)^{-1})U(\phi U + \zeta_y I)^{-1}\right] \tag{247}$$

$$= \mathrm{Tr}_d\left[U(\phi U + \zeta_y I)^{-1} - \frac{\zeta_x}{\zeta_y - \zeta_x}(U(\phi U + \zeta_x I)^{-1} - U(\phi U + \zeta_y I)^{-1})\right] \tag{248}$$

$$= \frac{1}{\zeta_y - \zeta_x}\mathrm{Tr}_d\left[\zeta_y U(\phi U + \zeta_y I)^{-1} - \zeta_x U(\phi U + \zeta_x I)^{-1}\right] \tag{249}$$

$$= \frac{1}{\zeta_y - \zeta_x}\left(\zeta_y + y - \zeta_x - x\right) \tag{250}$$

$$= 1 + \frac{y - x}{\zeta_y - \zeta_x} \tag{251}$$

Hence the general formula:

$$\kappa_2 = -\frac{\zeta_y - \zeta_x}{y - x} \tag{252}$$

One can check that the same formula applies for instance for the mismatched ridgeless regression. Also, we assume that it can be replaced by its continuous limit in $y \to x$ in the situation $x = y$.

Finally for $f_2$, we find

$$f_2 = c_0 - \mathrm{Tr}_d\left[\zeta_z V(\phi U + \zeta_z I)^{-1}\right] \tag{253}$$

$$= c_0 - \mathrm{Tr}_d\left[\zeta_z V_{F\perp}(\phi U_{F\perp} + \zeta_z I)^{-1}\right] - \mathrm{Tr}_d\left[\zeta_z V_F(\phi U_F + \zeta_z I)^{-1}\right] \tag{254}$$

$$= c_0 - \sigma^2 - \lim_{d \to +\infty}\left(\frac{p}{d}\tilde{g}^{\langle 11\rangle} + \frac{N}{d}\tilde{g}^{\langle 22\rangle}\right) \tag{255}$$

$$= c_0 - \sigma^2 - \psi(\tilde{g}^{\langle 11\rangle} + \phi_0\tilde{g}^{\langle 22\rangle}) \tag{256}$$

where we use $\tilde{g}$ associated to a slightly different linear-pencil $\tilde{M}$:

$$\tilde{M} = \begin{pmatrix} 0 & 0 & I & 0 \\ 0 & 0 & 0 & I \\ \zeta_z I & A_F & -\zeta_z V_F & 0 \\ A_F^T & -\frac{1}{\phi}I & 0 & 0 \end{pmatrix} \tag{257}$$

from which we get using a Compute-Algebra-System

$$\psi\tilde{g}^{\langle 11\rangle} + \psi\phi_0\tilde{g}^{\langle 22\rangle} = r^2\delta_z \tag{258}$$

Another more straightforward way for obtaining the same result without the need for an additional linear-pencil is to notice that if we let $E_1 = (I_p|0_{p\times N})$ such that $V_F = \tilde{r}^2 E_1 E_1^T$, then we have:

$$\mathrm{Tr}_d\left[\zeta_x V_F(\phi U_F + \zeta_x I)^{-1}\right] = \mathrm{Tr}_d\left[\zeta_x\tilde{r}^2 E_1^T(\phi U_F + \zeta_x I)^{-1}E_1\right] \tag{259}$$

$$= \tilde{r}^2\zeta_x\mathrm{Tr}_p\left[E_1^T(\phi U_F + \zeta_x I)^{-1}E_1\right] \tag{260}$$

Therefore reusing the definition of $\delta_x$ and the former linear-pencil $M$:

$$\mathrm{Tr}_d\left[\zeta_x V_F(\phi U_F + \zeta_x I)^{-1}\right] = \tilde{r}^2\psi\zeta_x g^{\langle 14\rangle} = r^2\delta_x \tag{261}$$

**Conclusion** we have the following equations

$$\tilde{f}_1(x, y) = \left(-\frac{\zeta_y - \zeta_x}{y - x}\right)\left(r^2\frac{1 + \nu^2\mu^2\psi_0(\gamma_x\gamma_y)}{1 - \mu^4\psi_0(\delta_x\delta_y)(\gamma_x\gamma_y)}(\delta_x\delta_y) + \sigma^2\right) \tag{262}$$

$$f_2(z) = c_0 - (r^2\delta_z + \sigma^2) \tag{263}$$

$$\delta_z = (1 + \gamma_z\mu^2\psi_0)^{-1} \tag{264}$$

$$\gamma_z = (\mu^2\delta_z + \phi_0^{-1}\zeta_z + \nu^2)^{-1} \tag{265}$$

$$\frac{\gamma_y}{\phi_0}\zeta_y = 1 - \frac{\phi_0}{\psi_0}\left(1 + \frac{y}{\zeta_y}\right) \tag{266}$$

### D.5 REALISTIC DATASETS

For the realistic datasets, we capture the time evolution for two different datasets: MNIST and Fashion-MNIST. To capture the dynamics over a realistic dataset $X \in \mathbb{R}^{n_{\text{tot}} \times d}$, it is more convenient to use the dual matrices $U_\star, V_\star, \Xi$. We only need to estimate $U_\star$ and $\Xi\beta^*$ with $U_\star \simeq \frac{1}{n_{\text{tot}}} X^T X$ and $\Xi\beta^* \simeq \frac{1}{n_{\text{tot}}} X^T Y$. In both cases, we sill sample a subset of $n < n_{\text{tot}}$ data-samples for the training set. The scope of the theoretical equations is still subject to the high-dimensional limit assumption, in other words we need $n$ and $d$ "large enough", that is to say $1 \ll n$. At the same time, the approximation of $U_\star$ and $\Xi\beta^*$ hints at $n_{\text{tot}}$ sufficiently large compared to the number of considered samples $n$. Hence we need also $n \ll n_{\text{tot}}$.

Numerically, for the two following datasets and as per assumptions 2.1, the theoretical prediction rely on a contour enclosing the spectrum $\text{Sp}(\tilde{X}^T \tilde{X})$ of $\tilde{X}^T \tilde{X}$, but not enclosing $-\lambda$. Therefore, in order to proceed with our computations, we take a symmetric rectangle around the x-axis crossing the axis at the particular values $-\frac{\lambda}{2}$ and $1.2 \max \text{Sp}(\tilde{X}^T \tilde{X})$ after a preliminary computation of the spectrum. For the need of our experiments, we commonly discretized the contour and ran a numerical integration over the discretized set of points.

**MNIST Dataset:** we consider the MNIST dataset with $n_{\text{tot}} = 70'000$ images of size $28 \times 28$ of numbers between 0 and 9. In our setting, we consider the problem of estimating the parity of the number, that is the vector $Y$ with $Y_i = 1$ if image $i$ represents an even number and $Y_i = -1$ for an odd-number. The dataset $X \in \mathbb{R}^{n_{\text{tot}} \times d}$ is further processed by centering each column to its mean, and normalized by the global standard-deviation of $X$ (in other words the standard deviation of $X$ seen as a flattened $n_{\text{tot}} \times d$ vector) and further by $\sqrt{d}$ (for consistency with the theoretical random matrix $Z$).

The results that we obtain are shown in Figure 4. On the figure on the left side we show the theoretical prediction of the training and test error with the minimum least-squares estimator (or alternatively the limiting errors at $t = +\infty$). We make the following observations which in fact relates to the same ones as in Figure 4 in Loureiro et al. (2021):

- There is an apparent larger deviation in the test error for smaller $n$ which tends to heal with increasing number of data samples

- A bias between the mean observation of the test error and the theoretical prediction emerges around the double-descent peak between $n = 100$ and $n = 1000$, in particular, the experiments are slightly above the given prediction. We notice that this bias is even more pronounced for smaller values of $\lambda$.

- Although it is not visible on the figure, increasing $n$ further tends to create another divergence between the theoretical prediction and the experimental runs - as it is expected with $n$ getting closer to $n_{\text{tot}}$.

Besides the limiting error, we chose to draw the time-evolution of the training and test error around at $n = 700$ around the double descent on the right side of Figure 4. This time, a gradient descent algorithm is executed for each 10 experimental runs with a constant learning-rate $dt = 0.01$. Due to the log-scale of the axis, it is interesting to notice that with such a basic non-adaptive learning-rate, each tick on the graph entails 10 times more computational time to update the weights. By contrast, the theoretical curves can be calculated at any point in time much farther away. Overall we see a good agreement between the evolution of the experimental runs with the theoretical predictions. However, as it is expected around the double-descent spike, learning-curves of the experimental runs appear slightly biased and above the theoretical curves.

**Fashion-MNIST Dataset:** We provide another example with MNIST-Fashion dataset with $d = 784$ and $n_{\text{tot}} = 70'000$. The dataset $X$ is processed as for the MNIST dataset. We take the output vector $Y$ such that $Y_i = 1$ for items $i$ above the waist, and $Y_i = -1$ otherwise. We provide the results in Figure 7 where the training set is sampled randomly with $n$ elements in $n_{\text{tot}}$ and the test set is sampled in the remaining examples. As it can be seen, the test error is slightly above the prediction for $n < 10^3$ but fits well with the predicted values for larger $n$. Furthermore, the learning curves through time in Figure 8 are different compared to the MNIST dataset in Figure 4 and we still observe a good match with the theoretical predictions. However the mismatch in the learning curves seems to increase in the specific case when $\lambda$ is lower, increasing thereby the effect of the double descent.

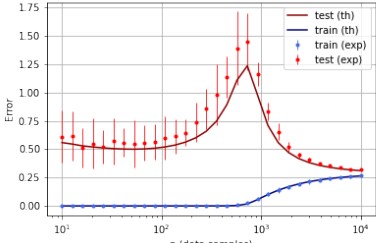 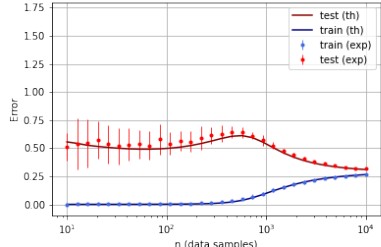

Figure 7: Comparison between the analytical and experimental learning profiles for the minimum least-squares estimator at $\lambda = 10^{-3}$ on the left (average and $\pm$ 2-standard-deviations over 20 runs) and $\lambda = 10^{-2}, n = 700$ on the right.

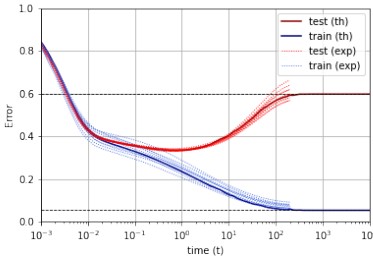

Figure 8: Comparison between the analytical and experimental learning evolution at $\lambda = 10^{-2}, n = 700$ (10 runs).

## E  LINEAR-PENCILS FIXED POINT EQUATION

In general, traces of algebraic expressions of large random matrices can be difficult to compute (See for instance appendix in Pennington & Worah (2017)). A modern approach consists in assembling a block of large random matrices such that the block-inversion formula (otherwise called the Schur complement) yields the desired algebraic expression of random matrices in some sub-blocks. Then, using the correlation structure of the sub-blocks of the assembled block-matrix, a fixed-point equation can be derived that yields a set of algebraic equations whose solutions provide the traces of the sub-blocks of the inverted block-matrix. This idea initially emerged in Rashidi Far et al. (2006), then has been described further in Mingo & Speicher (2017) and Helton et al. (2018) which coined the term "Linear-Pencils". Since then, it has recently been introduced in the machine learning community in a more geneal form in Adlam & Pennington (2020a), and also recently in Bodin & Macris (2021) where a non-rigorous proof is provided using the replica symmetry tool from statistical physics.

Here we propose to generalize even further the fixed-point equation where we let the sub-blocks be potentially of any form, and provide a non-rigorous proof of the proposition following the steps proposed in Bun et al. (2017); Potters & Bouchaud (2020) using Dyson brownian motions and Itô Lemma to derive the fixed-point equation.

### E.1  NOTATIONS AND MAIN STATEMENT

Let's consider an invertible self-adjoint complex block matrix $M \in \mathbb{C}^{N \times N}$ with $N = p_1 + \ldots + p_n$ such that $M^{\langle ij \rangle}$ is the sub-matrix of size $p_i \times p_j$. We assume that $p_1, \ldots, p_n \to \infty$ when $N \to \infty$ such that we have the fixed ratios $\gamma_i = \lim_{N \to \infty} \frac{p_i}{N}$, and let's define the inverse $G = M^{-1}$.

Now let $\mathbb{S} = \{ij | p_i = p_j\}$. We define for $(ij) \in \mathbb{S}$ (and defined to 0 outside of this set):

$$g_N^{\langle ij \rangle} = \frac{1}{p_i} \mathrm{Tr} \left[ G^{\langle ij \rangle} \right] = \frac{1}{p_i} \sum_{k=1}^{p_i} G_{kk}^{\langle ij \rangle} \qquad (267)$$

We further decompose $M$ as the sum of two components $M = M_0 + \frac{1}{\sqrt{N}} H$ with $M_0$ and $H$ both self-adjoint, $M_0$ is also invertible and where $H$ is a block of random matrices independent of $M_0$. In particular, $\mathrm{Re}(H)$ and $\mathrm{Im}(H)$ are independent element-wise with each-other, and we leave the possibility that the sub-blocks of $\mathrm{Re}(H)$ and $\mathrm{Im}(H)$ be either a Wigner random-matrix, Wishart random matrix, the adjoint of a Wishart random matrix, or a (real-)weighted sum of any of the three. For the sake of simplicity, we will consider that the elements $\mathrm{Re}\, H_{uv}^{\langle ij\rangle}$ or $\mathrm{Im}\, H_{uv}^{\langle ij\rangle}$ within the block $ij$ are gaussian and identically distributed although the gaussian assumption can certainly be weakened.

Now let's define $\sigma_{ij}^{kl}$ the covariance between the elements of the sub-matrices $H^{\langle ij\rangle}$ and $H^{\langle kl\rangle}$, that is for $(il, jk) \in \mathbb{S}^2$ on the off-diagonal position $uv$ on transposed element-locations:

$$\sigma_{ij}^{kl} = \mathbb{E}\left[H_{uv}^{\langle ij\rangle} H_{vu}^{\langle kl\rangle}\right] = \mathbb{E}\left[\mathrm{Re}\, H_{uv}^{\langle ij\rangle} \mathrm{Re}\, H_{vu}^{\langle kl\rangle}\right] - \mathbb{E}\left[\mathrm{Im}\, H_{uv}^{\langle ij\rangle} \mathrm{Im}\, H_{vu}^{\langle kl\rangle}\right] \tag{268}$$

Also, there can be some covariances on similar element-locations which we define with $\bar\sigma$ for $(ik, jl) \in \mathbb{S}^2$:

$$\bar\sigma_{ij}^{kl} = \mathbb{E}\left[H_{uv}^{\langle ij\rangle} H_{uv}^{\langle lk\rangle}\right] = \mathbb{E}\left[H_{uv}^{\langle ij\rangle} \bar H_{vu}^{\langle kl\rangle}\right] \tag{269}$$

$$= \mathbb{E}\left[\mathrm{Re}\, H_{uv}^{\langle ij\rangle} \mathrm{Re}\, H_{vu}^{\langle kl\rangle}\right] + \mathbb{E}\left[\mathrm{Im}\, H_{uv}^{\langle ij\rangle} \mathrm{Im}\, H_{vu}^{\langle kl\rangle}\right] \tag{270}$$

Notice that $\sigma_{ij}^{kl} = \sigma_{kl}^{ij}$ and $\bar\sigma_{ij}^{kl} = \bar\sigma_{lk}^{ji}$ by symmetry, and also when $H$ is real, we always have $\sigma_{ij}^{kl} = \bar\sigma_{ij}^{kl}$. So overall the random matrix $H$ has to satisfy the following property at any off-diagonal locations $(uv), (xy)$ and blocks $(ij, kl)$:

$$\delta_{\mathbb{S}^2}(jk, il)\delta_{vx}\delta_{yu}\sigma_{ij}^{kl} + \delta_{\mathbb{S}^2}(ik, jl)\delta_{ux}\delta_{vy}\bar\sigma_{ij}^{lk} = \mathbb{E}\left[H_{uv}^{\langle ij\rangle} H_{xy}^{\langle kl\rangle}\right] \tag{271}$$

Finally we define the mapping $\eta : \mathbb{C}^{n\times n} \to \mathbb{C}^{n\times n}$:

$$[\eta(g)]_{ij} = \sum_{kl\in\mathbb{S}} \gamma_k \sigma_{ik}^{lj} g^{\langle kl\rangle} \tag{272}$$

then let $\Pi(M) = M_0 - \eta(g) \otimes I$ with the notation $(\eta(g) \otimes I)_{ij} = \eta(g)_{ij} I_{p_i}$ when $ij \in \mathbb{S}$ and $(\eta(g) \otimes I)_{ij} = 0_{p_i \times q_j}$ the null-matrix when $ij \notin \mathbb{S}$. Similarly as $G$, with $\Pi(G) = \Pi(M)^{-1}$ and with:

$$g^{\langle ij\rangle} := \lim_{N\to\infty} g_N^{\langle ij\rangle} \tag{273}$$

$$\mathrm{Tr}_{p_i}\left[G^{\langle ij\rangle}\right] := \lim_{p_i\to\infty} \frac{1}{p_i}\mathrm{Tr}\left[G^{\langle ij\rangle}\right] \tag{274}$$

we state that:

$$g^{\langle ij\rangle} = \mathrm{Tr}_{p_i}\left[\Pi(G)^{\langle ij\rangle}\right] \tag{275}$$

**Remark 1:** When $M_0 = Z \otimes I$ such that $Z_{ij} = 0$ if $ij \notin \mathbb{S}$, then we get $\Pi(M) = (Z - \eta(g)) \otimes I$, then $\Pi(M)^{-1} = (Z - \eta(g))^{-1} \otimes I$. Therefore: $g = (Z - \eta(g))^{-1}$, or re-adjusting the terms, we find back the equation from Adlam & Pennington (2020a); Bodin & Macris (2021):

$$Zg = I_n + \eta(g)g \tag{276}$$

**Remark 2:** When considering the linear pencil of a block-matrix $M_\star$ such this is not necessarily self-adjoint however still invertible, the amplified matrix $M$ can be considered:

$$M = \begin{pmatrix} 0 & M_\star \\ \bar M_\star^T & 0 \end{pmatrix} \tag{277}$$

This implies that:

$$M^{-1} = \begin{pmatrix} 0 & (\bar M_\star^T)^{-1} \\ M_\star^{-1} & 0 \end{pmatrix} \tag{278}$$

So $g$ will also be of the form:

$$g = \begin{pmatrix} 0 & \bar g_\star^T \\ g_\star & 0 \end{pmatrix} \qquad \eta(g) = \begin{pmatrix} 0 & \bar\eta(g_\star)^T \\ \eta(g_\star) & 0 \end{pmatrix} \tag{279}$$

So in fact, the same equation still holds with $g_\star^{\langle ij\rangle} = \mathrm{Tr}_{p_i}\left[(C_\star - \eta(g_\star) \otimes I)^{\langle ij\rangle}\right]$ and thus, the self-adjoint constraints can be relaxed.

### E.2 Non-rigorous proof via Dyson brownian motions

In order to show the former result, we extend the sketch of proof provided in Bun et al. (2017); Potters & Bouchaud (2020). First we introduce a time $t$ and a matrix $Z \in \mathbb{C}^{n \times n}$ with $M(t, Z) = Z \otimes I + M_0 + \frac{1}{\sqrt{N}} H(t)$ with $H$ a Dyson brownian motion. Therefore, the matrix that we are interested in is actually $M = M(1, 0_{n \times n})$. In order for $H(1)$ to satisfy the property 271 we must have:

$$\mathrm{d}\left[H_{uv}^{\langle ij \rangle}, H_{xy}^{\langle kl \rangle}\right]_t = (\delta_{\mathbb{S}^2}(ik, jl)\delta_{ux}\delta_{vy}\bar{\sigma}_{ij}^{lk} + \delta_{\mathbb{S}^2}(il, jk)\delta_{uy}\delta_{vx}\sigma_{ij}^{kl})\mathrm{d}t \tag{280}$$

Itô's lemma provides the stochastic differential equation

$$\mathrm{d}G_{pq}^{\langle \alpha\beta \rangle} = \sum_{ij}\sum_{uv}\frac{\partial G_{pq}^{\langle \alpha\beta \rangle}}{\partial M_{uv}^{\langle ij \rangle}}\mathrm{d}M_{uv}^{\langle ij \rangle} + \frac{1}{2}\sum_{ijkl}\sum_{uvxy}\frac{\partial G_{pq}^{\langle \alpha\beta \rangle}}{\partial M_{uv}^{\langle ij \rangle}\partial M_{xy}^{\langle kl \rangle}}\mathrm{d}[M_{uv}^{\langle ij \rangle}, M_{xy}^{\langle kl \rangle}] \tag{281}$$

Using simple algebraic manipulations and the fact that $G$ is analytic in $M_{uv}^{\langle ij \rangle}$ as a rational function, we can rewrite the above partial derivatives as:

$$\frac{\partial G_{pq}^{\langle \alpha\beta \rangle}}{\partial M_{uv}^{\langle ij \rangle}} = -\left[G\frac{\partial M}{\partial M_{uv}^{\langle ij \rangle}}G\right]_{pq}^{\langle \alpha\beta \rangle} = -G_{pu}^{\langle \alpha i \rangle}G_{vq}^{\langle j\beta \rangle} \tag{282}$$

And applying the same formula twice:

$$\frac{\partial G_{pq}^{\langle \alpha\beta \rangle}}{\partial M_{uv}^{\langle ij \rangle}\partial M_{xy}^{\langle kl \rangle}} = G_{px}^{\langle \alpha k \rangle}G_{yu}^{\langle li \rangle}G_{vq}^{\langle j\beta \rangle} + G_{pu}^{\langle \alpha i \rangle}G_{vx}^{\langle jk \rangle}G_{yq}^{\langle l\beta \rangle} \tag{283}$$

Injecting it in (281) we get for $p = q$

$$\mathrm{d}G_{pp}^{\langle \alpha\beta \rangle} = -\frac{1}{\sqrt{N}}\sum_{ij}\sum_{uv}G_{pu}^{\langle \alpha i \rangle}G_{vp}^{\langle j\beta \rangle}\mathrm{d}H_{uv}^{\langle ij \rangle} \tag{284}$$

$$+ \frac{1}{2N}\sum_{(ik,jl)\in\mathbb{S}^2}\sum_{uv}\left[G_{pu}^{\langle \alpha k \rangle}G_{vu}^{\langle li \rangle}G_{vp}^{\langle j\beta \rangle} + G_{pu}^{\langle \alpha i \rangle}G_{vu}^{\langle jk \rangle}G_{vp}^{\langle l\beta \rangle}\right]\bar{\sigma}_{ij}^{lk}\mathrm{d}t \tag{285}$$

$$+ \frac{1}{2N}\sum_{(il,jk)\in\mathbb{S}^2}\sum_{uv}\left[G_{pv}^{\langle \alpha k \rangle}G_{uu}^{\langle li \rangle}G_{vp}^{\langle j\beta \rangle} + G_{pu}^{\langle \alpha i \rangle}G_{vv}^{\langle jk \rangle}G_{up}^{\langle l\beta \rangle}\right]\sigma_{ij}^{kl}\mathrm{d}t \tag{286}$$

So considering $\gamma_\alpha \mathrm{d}g_N^{\langle \alpha\beta \rangle} = \frac{1}{N}\mathrm{d}\sum_p G_{pp}^{\langle \alpha\beta \rangle}$ we get

$$\gamma_\alpha \mathrm{d}g_N^{\langle \alpha\beta \rangle} = \epsilon_N^{\langle \alpha\beta \rangle} + \frac{1}{2N}\sum_{(il,jk)\in\mathbb{S}^2}\sum_p\left[[G^{\langle \alpha k \rangle}G^{\langle j\beta \rangle}]_{pp}\gamma_l g_N^{\langle li \rangle} + [G^{\langle \alpha i \rangle}G^{\langle l\beta \rangle}]_{pp}\gamma_j g_N^{\langle jk \rangle}\right]\sigma_{ij}^{kl}\mathrm{d}t \tag{287}$$

where:

$$\epsilon_N^{\langle \alpha\beta \rangle} = -\frac{1}{N^{\frac{3}{2}}}\sum_{ij}\sum_{uv}\sum_p G_{pu}^{\langle \alpha i \rangle}G_{vp}^{\langle j\beta \rangle}\mathrm{d}H_{uv}^{\langle ij \rangle} \tag{288}$$

$$+ \frac{1}{2N^2}\sum_{(ik,jl)\in\mathbb{S}^2}\sum_{uv}\sum_p\left[G_{pu}^{\langle \alpha k \rangle}G_{vu}^{\langle li \rangle}G_{vp}^{\langle j\beta \rangle} + G_{pu}^{\langle \alpha i \rangle}G_{vu}^{\langle jk \rangle}G_{vp}^{\langle l\beta \rangle}\right]\bar{\sigma}_{ij}^{lk}\mathrm{d}t \tag{289}$$

The matrix $Z$ is now helpful upon noticing that (using again the analyticity of $G$)

$$\frac{\partial G_{pp}^{\langle \alpha\beta \rangle}}{\partial Z_{kj}} = -\left[G\frac{\partial M}{\partial Z_{kj}}G\right]_{pp}^{\langle \alpha\beta \rangle} = -[G(E_{kj} \otimes I)G]_{pp}^{\langle \alpha\beta \rangle} = -\left[G^{\langle \alpha k \rangle}G^{\langle j\beta \rangle}\right]_{pp} \tag{290}$$

Hence (using the fact that $\sigma_{ij}^{kl} = \sigma_{kl}^{ij}$)

$$\gamma_\alpha \mathrm{d}g_N^{\langle\alpha\beta\rangle} = \epsilon_N^{\langle\alpha\beta\rangle} - \frac{1}{2N} \sum_{(il,jk)\in\mathbb{S}^2} \sum_p \left[ \frac{\partial G_{pp}^{\langle\alpha\beta\rangle}}{\partial Z_{kj}} \gamma_l g_N^{\langle li\rangle} + \frac{\partial G_{pp}^{\langle\alpha\beta\rangle}}{\partial Z_{il}} \gamma_j g_N^{\langle jk\rangle} \right] \sigma_{ij}^{kl} \mathrm{d}t \tag{291}$$

$$= \epsilon_N^{\langle\alpha\beta\rangle} - \frac{\gamma_\alpha}{2} \sum_{(il,jk)\in\mathbb{S}^2} \left[ \sigma_{kl}^{ij} \gamma_l g_N^{\langle li\rangle} \frac{\partial g_N^{\langle\alpha\beta\rangle}}{\partial Z_{kj}} + \sigma_{ij}^{kl} \gamma_j g_N^{\langle jk\rangle} \frac{\partial g_N^{\langle\alpha\beta\rangle}}{\partial Z_{il}} \right] \mathrm{d}t \tag{292}$$

$$= \epsilon_N^{\langle\alpha\beta\rangle} - \frac{\gamma_\alpha}{2} \left[ \sum_{jk\in\mathbb{S}} [\eta(g_N)]_{kj} \frac{\partial g_N^{\langle\alpha\beta\rangle}}{\partial Z_{kj}} + \sum_{il\in\mathbb{S}} [\eta(g_N)]_{il} \frac{\partial g_N^{\langle\alpha\beta\rangle}}{\partial Z_{il}} \right] \mathrm{d}t \tag{293}$$

$$= \epsilon_N^{\langle\alpha\beta\rangle} - \gamma_\alpha \left[ \sum_{jk\in\mathbb{S}} [\eta(g_N)]_{kj} \frac{\partial g_N^{\langle\alpha\beta\rangle}}{\partial Z_{jk}} \right] \mathrm{d}t \tag{294}$$

As it would require more in-depth analysis, we make the following two assumptions:

1. We assume that $g_N^{\langle\alpha\beta\rangle}$ concentrates towards a constant value $g^{\langle\alpha\beta\rangle}$ when $N \to \infty$

2. That $\epsilon_N^{\langle\alpha\beta\rangle}$ concentrates towards 0 when $N \to \infty$

With these assumptions in mind we obtain the partial differential equation:

$$\frac{\partial g^{\langle\alpha\beta\rangle}}{\partial t} + \sum_{ij\in\mathbb{S}} [\eta(g)]_{ij} \frac{\partial g^{\langle\alpha\beta\rangle}}{\partial Z_{ij}} = 0 \tag{295}$$

Finally, using the change of variable $\hat{g}(s) = g(\hat{t}(s), \hat{Z}(s)) = g(t+s, Z+s\eta(g(t,Z)))$ we find:

$$\frac{\mathrm{d}\hat{g}^{\langle\alpha\beta\rangle}}{\mathrm{d}s} = \frac{\partial g^{\langle\alpha\beta\rangle}}{\partial t} \frac{\partial \hat{t}}{\partial s} + \sum_{ij} \frac{\partial g^{\langle\alpha\beta\rangle}}{\partial Z_{ij}} \frac{\partial \hat{Z}_{ij}}{\partial s} \tag{296}$$

$$= \frac{\partial g^{\langle\alpha\beta\rangle}}{\partial t} + \sum_{ij} \frac{\partial g^{\langle\alpha\beta\rangle}}{\partial Z_{ij}} [\eta(g(t,Z))]_{ij} = 0 \tag{297}$$

So $\hat{g}^{\langle\alpha\beta\rangle}(s)$ is constant so: $\hat{g}^{\langle\alpha\beta\rangle}(-t) = \hat{g}^{\langle\alpha\beta\rangle}(0)$ which implies:

$$g(0, Z - t\eta(g(t,Z))) = g(t, Z) \tag{298}$$

Hence for $(t, Z) = (1, 0_{n\times n})$ we have:

$$g(0, -\eta(g(1,0))) = g(1,0) \tag{299}$$

Hence the expected result:

$$g^{\langle ij\rangle} = [g(1,0)]_{ij} = [g(0, -\eta(g(1,0)))]_{ij} = \mathrm{Tr}_{p_i} \left[ \left[ (M_0 - \eta(g(1,0)) \otimes I)^{-1} \right]^{\langle ij\rangle} \right] = \mathrm{Tr}_{p_i} \left[ \Pi(G)^{\langle ij\rangle} \right] \tag{300}$$

### E.3 EXAMPLES

#### WIGNER SEMICIRCLE LAW

Let's consider $n = 1$ and the symmetric random matrix $H \in \mathbb{R}^N$ and $M = \frac{H}{\sqrt{N}} - zI$. We find that $\eta(g) = g$ and using (276) we find directly

$$-zg = 1 + g^2 \tag{301}$$

MARCHENKO PASTUR LAW

Let's consider $n = 2$ and the random matrix $X \in \mathbb{R}^{d \times N}$ with $\phi = \frac{N}{d} = \frac{\gamma_1}{\gamma_2}$ and $\gamma_1 = \frac{N}{N+d}, \gamma_2 = \frac{d}{N+d}$ and the random symmetric block matrix:

$$M = \begin{pmatrix} -zI_N & \frac{X^T}{\sqrt{N}} \\ \frac{X}{\sqrt{N}} & -I_d \end{pmatrix} = \begin{pmatrix} -zI_N & \frac{1}{\sqrt{\gamma_1}}\frac{X^T}{\sqrt{N+d}} \\ \frac{1}{\sqrt{\gamma_1}}\frac{X}{\sqrt{N+d}} & -I_d \end{pmatrix} \tag{302}$$

Using Schur complement, it can be seen that $g_N^{\langle 11 \rangle} = \frac{1}{N}\mathrm{Tr}\left[\left(\frac{X^TX}{N} - zI_N\right)^{-1}\right]$ which is precisely the trace that is being looked for.

A careful analysis shows that $\sigma_{12}^{12} = \sigma_{21}^{21} = \frac{1}{\gamma_1}$ while the rest is null and thus $[\eta(g)]_{11} = \frac{\gamma_2}{\gamma_1}g^{\langle 22 \rangle} = \frac{g^{\langle 22 \rangle}}{\phi}$ and $[\eta(g)]_{22} = g^{\langle 11 \rangle}$. With (276) we obtain the system of algebraic equations

$$-zg^{\langle 11 \rangle} = 1 + \frac{1}{\phi}g^{\langle 22 \rangle}g^{\langle 11 \rangle} \tag{303}$$

$$-g^{\langle 22 \rangle} = 1 + g^{\langle 11 \rangle}g^{\langle 22 \rangle} \tag{304}$$

Therefore, injecting the solution of the second equation $g^{\langle 22 \rangle} = -\frac{1}{1+g^{\langle 11 \rangle}}$ in the first equation

$$-zg^{\langle 11 \rangle} = 1 - \frac{1}{\phi}\frac{g^{\langle 11 \rangle}}{1 + g^{\langle 11 \rangle}} \tag{305}$$

Hence the Marcheko Pastur result:

$$z(g^{\langle 11 \rangle})^2 + \left(z + 1 - \frac{1}{\phi}\right)g^{\langle 11 \rangle} + 1 = 0 \tag{306}$$

