# OpenReview forum: "Gradient flow in the gaussian covariate model: exact solution of learning curves and multiple descent structures"
_ICLR.cc/2023/Conference — Submitted to ICLR 2023_

### Official Review · Reviewer_amF8 · 2022-10-21

**Confidence:** 4
**Correctness:** 4
**Technical Novelty And Significance:** 3
**Empirical Novelty And Significance:** 3
**Recommendation:** 6

**Clarity, Quality, Novelty And Reproducibility:**

- It would help my review if the authors could comment on what is completely novel in this work. My current sense is that the main contribution is to generalize previous work, including Bodin and Macris, to the full Gaussian covariates setting.
- Just after Eq (8), the assumption that U and V* commute is mentioned to provide a limiting self-consistent equation. Can this be avoided by introducing a joint spectral density like Assumption 1 in [1]?
- In Sec. 1.1, the authors should consider citing [2] and [3] in relation to the analytic calculation of double-descent curves. Similarly [4] is a relevant citation from triple or even multiple descents for the generalization error.
- I think Eq (9) has a typo since there is the matrix I in the denominator.

[1] Tripuraneni, Nilesh, Ben Adlam, and Jeffrey Pennington. "Overparameterization improves robustness to covariate shift in high dimensions." Advances in Neural Information Processing Systems 34 (2021): 13883-13897.

[2] Lin, Licong, and Edgar Dobriban. "What Causes the Test Error? Going Beyond Bias-Variance via ANOVA." J. Mach. Learn. Res. 22 (2021): 155-1.

[3] Adlam, Ben, and Jeffrey Pennington. "Understanding double descent requires a fine-grained bias-variance decomposition." Advances in neural information processing systems 33 (2020): 11022-11032.

[4] Adlam, Ben, and Jeffrey Pennington. "The neural tangent kernel in high dimensions: Triple descent and a multi-scale theory of generalization." International Conference on Machine Learning. PMLR, 2020.

**Strength And Weaknesses:**

**Strengths**

- The setting of Gaussian covariates is nice and generalizes many other models.
- Early stopping is important in practice. This theory gives some results on how to choose the stopping to avoid multiple-descent peaks.
- The analysis connecting the peaks in Fig 2 to the spectrum in Fig 3 is nice.
- The use of Dyson’s Brownian motion to derive the self-consistent equation (276) is cool.


**Weaknesses**

- No significant new machine learning implications from the results are derived in the paper. This might limit the paper’s interest to the wider ICLR community.
- I couldn’t find any details of how numerical predictions are generated from the equations of Result 2.1. In particular, how are the contour integrals calculated? Which contour is used? How do you avoid spurious solutions to the self-consistent equation?

**Summary Of The Paper:**

The authors study the evolution under gradient flow of the training and test errors in the setting of Gaussian covariate models, which encompasses several other well studied settings. Using tools from random matrix theory, in particular the linear pencil, they calculate the exact asymptotic behavior of several quantities in the high-dimensional limit. The authors show how multiple descent phenomena can arise as a function of both number of parameters and training time. The paper also includes results on real datasets and shows good agreement between simulations and theoretical predictions.

**Summary Of The Review:**

The paper contains nice results that unify and extend previous work. The paper's main claims are supported by proofs and illustrative experiments. Overall the paper is clearly written and covers a lot of material.

---

> ### Author Response · Authors · 2022-11-18
> **Response to reviewer amF8**
>
> Dear reviewer, thank you first for all these observations and in particular for the typo that you mentioned and has been corrected, and the references that we added in the introduction (Note that reference 4 that you mentioned was already cited in 3 other locations). We also take good note of your remark about the joint-spectral density which is an interesting suggestion that we didn’t think about and must still investigate.
>
> The question related to how we choose the contour to calculate the integrals is quite relevant and we added some details in section D.5.
> Concerning the spurious solutions, this is actually a relevant point: finding the appropriate solution - in particular for zeta - is not an easy task and the literature remains elusive on that point (see the conditions in [1] or [4]). Selecting a correct branch of different solutions is in fact a problem in itself in random matrix literature (see lecture notes in [6]). For realistic datasets, in the same spirit than reference [2] with source code available in [5], using an iterative algorithm to find a fixed-point solution is enough to provide the correct solution. A possible consistency check is to verify if the value of zeta would match that of m(x) using the remark given at the end of B.1. However, for the anisotropic model for p=3 a set of 4 solutions exists for zeta(x) for each point x along the contour in the integration. In that specific case, we can resort to other tricks to select the appropriate solution. For instance according to D.2.1, x/zeta(x) has to match the eigenvalue distribution with a contour sufficiently close to the real axis. The appropriate solution can also be discriminated based on the sign of the imaginary part (which has to correspond to that of Im(x)), the symmetry along the x-axis and continuity along the contour.
>
> Regarding the more general concern about the contributions, we would like to refer to the other answer for reviewer utMN where we specify what we think is of interest for the research community with this analysis. In particular, we believe there is phenomenological value in the time-evolution of the generalization error. Indeed non-trivial time-dependent patterns emerge.
>
>
> [1] http://proceedings.mlr.press/v119/adlam20a.html
>
> [2] https://arxiv.org/pdf/2102.08127.pdf
>
> [4] https://arxiv.org/abs/cs/0610045
>
> [5] https://github.com/IdePHICS/GCMProject/blob/main/real_data/mnist_scattering.ipynb
>
> [6] https://www.cambridge.org/core/books/first-course-in-random-matrix-theory/2292A554A9BB9E2A4697C35BCE920304

---

### Official Review · Reviewer_utMN · 2022-10-25

**Confidence:** 3
**Correctness:** 4
**Technical Novelty And Significance:** 2
**Empirical Novelty And Significance:** 3
**Recommendation:** 6

**Clarity, Quality, Novelty And Reproducibility:**

I think the clarity of this paper can be improved if the authors give more discussion on the problem setting, intuition, and comparison to recent works. The novelty of this paper mainly lies in the observation of multiple descent with respect to the training epochs.

**Strength And Weaknesses:**

Strengths:

1.  Multiple descent and learning in high dimensions is an important and interesting topic. This paper proposes some timely and interesting results.

2. The paper provides rigorous proof of the generalization error in the gradient descent flow, and also provides some simulations and experiments.

Weaknesses:

My major concern is that there seem to be some confusing/vague statements. Below are some examples:

1. Given existing analyses on double/multiple descent in linear models and the regularization effect of early stopping, the technical contribution of the paper may be limited.

2. Recently, there are some papers related to multiple descent (arXiv:2204.10425, arXiv:2205.14846, arXiv:2208.09897). The connection between this paper and existing works is not discussed very thoroughly.

3. The theory applies to the high dimensional setting, so setting $p=3$ in the experiments may not be very reasonable. The authors should provide simulation results with larger $p$.

**Summary Of The Paper:**

This paper studies gradient descent for linear regression under the high dimensional framework. Under certain conditions, the authors study the generalization error in such gradient descent, and show that there exists multiple descent in the generalization error of the linear model during training. The authors also conduct experiments on the random feature models and realistic data sets.

**Summary Of The Review:**

In summary, this paper gives solid and interesting results. However, as discussed above, several parts of the paper can still be improved.

---

> ### Author Response · Authors · 2022-11-18
> **Response to reviewer utMN**
>
> Dear reviewer, thank you for your interesting remarks and references suggestions which we take into account.
>
> The papers that you suggest are interesting and present recent results. They provide new aspects into consideration with the analysis of different polynomial scalings of the aspect ratios. These new insights didn’t slip under our radar, however, we believe they are beyond the scope of our current analysis. It would definitely be an interesting endeavor to examine if there is a possibility to extend our analysis to these new scaling regimes, hence we mention this as an interesting open problem in the conclusion.
>
> Concerning the “p=3” problem, we believe the confusion stems from using the same notation for two different things. Indeed: p is a high-dimensional size of a matrix in the model description in section 1.2, while it is also defined as the finite-size for a “p x p” high-dimensional random block matrix in Figure 1 (see estimator matrix A in table 1, row 3). We are really sorry for the inconvenience and took your input into consideration (variable changed in 1.2).
>
> As also mentioned for reviewer RBfi and FVzp, we believe that besides the observation of the multiple descent with respect to the training epochs that you accurately mentioned, a few other aspects also emerge from this work:
>
> _ The approach to derive the equations rests solely on random matrix theory, using linear-pencils, and is quite recent. It differs from the one in [2]. We also contributed further to improve the tools described in [1] but unfortunately had to push this part in the appendix because of the lack of space (see appendix E). We believe that random matrix theory can have a more prominent role in understanding better large learning models and believe our analysis contributes in going in that direction.
>
> _ That it is even possible to derive analytical formulas, besides being somewhat remarkable, is also important because it enables predicting the test error and training error at all times without the burden of calculating a large vector of weights. Moreover there appears to be a good matching of these formulas with realistic datasets.
>
> [1] http://proceedings.mlr.press/v119/adlam20a.html
>
> [2] https://arxiv.org/pdf/2102.08127.pdf

---

### Official Review · Reviewer_RBfi · 2022-10-26

**Confidence:** 3
**Correctness:** 4
**Technical Novelty And Significance:** 3
**Empirical Novelty And Significance:** Not applicable
**Recommendation:** 6

**Clarity, Quality, Novelty And Reproducibility:**

**Clarity**
---

While the writing of the paper is clear, the equations and the results are fairly opaque.

**Novelty and Significance**
---

The analysis techniques are not the most novel. However, the generality of the setting and then derivation of known results in this unified setting is significant.

**Reproducibility**
---

The paper should be reproducible.

**Questions**
---

1) Can the authors provide some insight into the formulas in Result 2.1?

**Strength And Weaknesses:**

**Strengths**
---

1) The set up is fairly general and captures many existing models that have been analyzed.
2) The method provides the whole training curve and not just the limiting distribution.
3) The paper re-captures known results/
4) The idea of evolutions of the eigenvalues and the ``bulk'' in section 3.2 is interesting.

**Weaknesses**
---

1) I think the results are very hard to parse specially Result 2.1 and result 2.4. Further in result 2.1 the terms depend on equations $\tilde{f}$ which is very complicated term which itself terms on equation (8) which is a complicated implicit equation (both sides depend on the quantity). Hence while the formulas might be true, it is not clear how informative they are. **That is, do they actually tell us much more than $\beta^\lambda = (\hat{X}^T\hat{X} + \lambda I )^{−1} \hat{X}^T Y$ and then just writing down the test error.**



**Summary Of The Paper:**

This paper looks at the regularized linear regression where the data is generated using a Gaussian Covariate model and the regression targets are linear functions of the data. the paper starts by showing the that the setting is fairly general and captures many other set ups. In this set up, under some assumptions the paper then obtains a formula (albeit a complicated one) for the training and test error as time evolves.

Then they specialize to specific instances and re-discover known results from prior work.



**Summary Of The Review:**

In summary the paper is a nice theoretical contribution however, the insights that can be readily obtained seem lacking. I think significant work would be required to simply the formulas or some work should be done to help understand the formulas (at heuristically) would go a long way to improving the paper.

---

> ### Author Response · Authors · 2022-11-18
> **Response to reviewer RBfi**
>
>
> Dear reviewer, thank you first for your valuable inputs and relevant remarks.
>
> The equation (8) is indeed - in our opinion - correctly identified as the most complicated one, as being defined as an implicit equation. It is however not so surprising as these kinds of algebraic equations tend to arise in the random matrix world. However, we also find it quite intriguing/striking to have this term, as this can lead to expressions which cannot be tracked by an algebraic solution. For instance with equation (177), the variable p can be made as big as desired so that zeta is the solution of a high-order polynomial equation which does not necessarily have any algebraic solution. Thus, we expect this term to be the source of richer training/test-error curves which cannot be expressed as contour integrals of simple algebraic formulas.
>
> We agree that the elements of the equations in (2.1) are difficult to articulate in terms of  interpretable statistical objects or properties. However, we see other benefits from using them (we stressed this more in section 1.3):
>
> _ They provide a straightforward formula to draw the picture of the evolution at all times for different parameters, while running and averaging the results on a large system can require excessive computational resources (we thus added a more thorough discussion of figure 4 and also response to reviewer FVzp). This in turn can help speed-up discovering phenomenological observations such as the double epoch-wise bumps in Figure 2.
>
> _ Extending further on the results of [2], the predictive power of these models seem to reach the scope of some learning models with realistic datasets. Our hope is that developing these methods and introducing tools using random matrix theory (such as the one that is described in Appendix E) could help us highlight undesirable regimes (such as around the double descent spike) or on the contrary illustrate behaviors that challenge the intuition (such as the benefit of overparameterization with its tradeoff with training steps). The ultimate goal would be to be able to tackle even larger class of models and potentially predict proper parameter tuning early on - before running any training algorithm.
>
>
> [2] https://arxiv.org/pdf/2102.08127.pdf

---

> > ### Comment · Reviewer_RBfi · 2022-11-29
> > **Thanks**
> >
> > Thanks for responding to my concerns.
> >
> > I am still a little concerned about the interpretability of the results. But I am still for accepting the paper.

---

### Official Review · Reviewer_FVzp · 2022-10-26

**Confidence:** 4
**Correctness:** 3
**Technical Novelty And Significance:** 3
**Empirical Novelty And Significance:** Not applicable
**Recommendation:** 5

**Clarity, Quality, Novelty And Reproducibility:**

**Clarity \& Quality:** the writing is mostly clear. I have the following questions:

- In the current manuscript there is no discussion around the assumptions, and it is not clear if they hold in the listed examples. For the random features example, I believe the activation function is zero-centered so that the spectral norm of the kernel matrix is asymptotically bounded. Can the authors confirm the conditions required for the activation?

- In the Appendix, I would appreciate some explanations on how the large block matrices are constructed. Do we know if the linear pencil is minimal in terms of the size?

- In Figure 4 (right), why are the experimental values only plotted until $t=10^2$? Does the test error start to deviate from the theoretical predictions after that?

**Novelty:** see weaknesses above.

**Reproducibility:** N/A.

**Strength And Weaknesses:**

## Strength

The studied problem is well motivated: most existing works on the precise asymptotics of training/test error only consider the empirical risk minimizer (instead of the entire gradient flow trajectory), and the time-evolution of the error reveals some interesting phenomena such as epoch-wise double descent. While the results are only derived for Gaussian features, we may expect wider applicability due to the Gaussian equivalence (universality) property. Moreover, the linear pencil construction and the Dyson Brownian motion derivation of the self-consistent equations could be useful in future random matrix-related research.

## Weaknesses

My main concern is that many relevant prior results are not discussed.

- Asymptotics of the least squares estimator in Section 3.2 (anisotropic features, isotropic teacher coefficients) was first rigorously studied in [Dobriban and Wager 2018]. This result was then extended to general teacher models using the Stieltjes transform approach in [Wu and Xu 2020] [Richards et al. 2020], where the authors already showed that the multiple descent risk curve can be engineered using eigenvalues of different scales. Also note that the self-consistent equations in these works are identical to the infinite time-limit in this submission.

- The discussion of the Gaussian equivalence principle in Section 3.3 is insufficient and non-rigorous. The cited [Peche 2019] only implies the (weak) equivalence in the kernel spectrum, which does not guarantee the equivalence in the training and test error. The authors should look into [Hu and Lu 2020] and references therein.

- Asymptotics of the gradient flow dynamics can also be analyzed using alternative approaches such as dynamical mean-field theory. In this submission, the squared loss gives a closed-form expression of the trajectory which simplifies the computation, but it would still be nice to briefly discuss and compare these different method.

Dobriban and Wager 2018. High-dimensional asymptotics of prediction: ridge regression and classification.
Wu and Xu 2020. On the optimal weighted $\ell_2$ regularization in overparameterized linear regression.
Richards et al. 2020. Asymptotics of ridge (less) regression under general source condition.
Hu and Lu 2020. Universality laws for high-dimensional learning with random features.

In addition, it is not entirely clear of how ML theory researchers can benefit from knowing these asymptotic formulae. Currently, the application seems to be limited to plotting good-looking risk curves; this does not provide quantitative characterization of the reported phenomena. For example, to rigorously establish epoch-wise double descent, the authors need to check the time derivative of the error to show non-monotonicity. Such analysis would strengthen the theoretical contribution of this submission.

**Summary Of The Paper:**

This submission computes the asymptotic training and test error of linear estimators optimized by gradient flow on the least squares objective, under the student-teacher setting with Gaussian covariates studied in [Loureiro et al. 2021]. The analysis generalizes the earlier linear pencil calculation for random features model in [Bodin and Macris 2021], and recovers the performance of the ridge regression estimator when the time $t$ is large. The authors present a few examples of solvable models, and also demonstrate empirically that the Gaussian covariate predictions can be accurate for certain real-world datasets.

**Summary Of The Review:**

My current evaluation is that this is a borderline submission: I believe the random matrix analysis is a solid contribution, but the authors need to thoroughly discuss prior works and present the results in a more accessible way for the ICLR community. I will consider updating my score if the authors can adequately address the above concerns.

---

> ### Author Response · Authors · 2022-11-18
> **Response to reviewer FVzp**
>
> Dear reviewer, thank you first for the additional references that we added in the relevant sections.
>
> We agree the use of the gaussian equivalence principle remains non-rigorous to apply it broadly as we did. This is however a common tool which is applied in similar problems as in [1] where 4 instances of the principle are applied to proceed with the linear-pencil calculation.
> Nevertheless, we agree we should stress more clearly that there is no complete rigorous foundation for this application. Thus, this along with the question regarding the zero-centering of the activation function led us to reformulate section 3.3 regarding the random feature model.
>
> Regarding Figure 4, there are two sides to distinguish and not be confused. Further details have been added in section D.5 and sum up to the following:
>
> _ The figure on the left side shows that there is a good agreement at t=infinity between the experimental and theoretical training and test errors, however, it can be seen that there is a slight bias around the double descent peak. This bias has already been seen in other datasets for instance in Figure 4 in [2]. The figure on the right side shows precisely this fact, although at a different lambda (here n=700 is around the peak): the experimental test error does appear to be slightly above the prediction, but not by far as can already be seen clearly at t=10^2
>
> _ Essentially, the reason why the experimental runs stop around t=10^2 while the theoretical one goes as far as t=10^4, is that the theoretical computation costs nothing compared to computing the experimental ones. Indeed: The 10 runs are launched for n=700 at a fixed learning rate dt=0.01 for 20’000 steps and took 8 minutes each (one could certainly improve the code or use better computational resources, but without changing anything, going one tick further on the graph would increase the time by 10 folds).
> This is also one of the main points of this analysis:  the theoretical computations are far easier to generate, in particular for the heatmaps, and can thus serve as precious design guidelines. We added a related remark in Contributions (sec 1.3).
>
> Constructing relevant linear-pencils seems to be a handcrafting exercise as far as we know. We do not know if we could potentially make them even simpler to derive the functions of interest. Here we have already a large improvement with at maximum a size 6x6 compared to the size 13x13 [3] and at the same time offer a larger class of models and even realistic datasets. Based on this remark, we add a short comment in appendix B.
>
> We also agree with you that taking the time derivative of the equation (3) and (4) of result 2.1 (in the same spirit as [7] section 2.2) can lead to fruitful observations, in particular for establishing optimal stopping times.
>
> Besides, although our results mainly concern random matrix theory applied with the gaussian covariate model, we agree with you that comparing this approach with DMFT is a worthwhile goal and mention this valuable remark in the conclusion for future work.
>
>
> [1] http://proceedings.mlr.press/v119/adlam20a.html
>
> [2] https://arxiv.org/pdf/2102.08127.pdf
>
> [3] https://arxiv.org/abs/2110.11805
>
> [4] https://arxiv.org/abs/cs/0610045
>
> [5] https://github.com/IdePHICS/GCMProject/blob/main/real_data/mnist_scattering.ipynb
>
> [6] https://www.cambridge.org/core/books/first-course-in-random-matrix-theory/2292A554A9BB9E2A4697C35BCE920304
>
> [7] https://www.sciencedirect.com/science/article/pii/S0893608020303117

---

### Decision · Program_Chairs · 2023-01-20

**Decision:**

Reject

**Justification For Why Not Higher Score:**

This paper presents non-trivial formulas that characterize the trajectory of gradient flow in the Gaussian covariate model. While these expressions have independent merit and may be interesting to parts of the theory community, the paper does little to draw scientific conclusions out of those formulas, and instead relies on their numerical evaluation to demonstrate relevant phenomenology. Given that the numerical evaluation is non-trivial, that insufficient details were provided about how to extract accurate numerical results, and that limited analytical conclusions were obtained without resorting to numerics, the contributions of this paper mainly rest on the presentation of the equations themselves, which is in itself insufficient for publication at ICLR.

**Justification For Why Not Lower Score:**

N/A.

**Metareview: Summary, Strengths And Weaknesses:**

This paper presents and analyzes formulas for the high-dimensional asymptotic trajectory of gradient flow for least-squares regression in the Gaussian covariate model. The results generalize those of Loureiro et al. (2021) to finite time and of Bodin & Macris (2021) to a broader class of models.

The reviewers appreciate the relatively general setup, the practical importance of studying early stopping, and the interesting phenomenology revealed through numerical investigation of the equations. The discussion highlighted several reviewer concerns, including insufficient discussion of prior work, lack of rigor in parts of the analysis, limited machine learning takeaways, and challenges with reproducibility.

The reviewers pointed out that the scope of the analysis becomes relatively limited without invoking universality/Gaussian equivalence, which has yet to be proven rigorously. The authors' response contends that it is a common tool which has been used in prior works without proof as well. Given that the consequence of forgoing the universality claims is merely a reduction in generality (rather than in invalidation of the results as a whole), I would agree with the authors that this limitation is not a significant weakness of the paper.

On the other hand, the reviewers also point out that the observed phenomenology and interpretation of the results is based entirely on numerical evaluation of the formulas, rather than an analytical investigation of the properties of the equations. This significantly reduces the value delivered by presenting the formulas in the first place, as it would be simpler and equally illuminating to perform a numerical evaluation of finite-sized models, which could be done without any theoretical analysis whatsoever.

Furthermore, the numerical evaluation of the equations is itself extremely non-trivial. One reviewer asked for clarification regarding the choice of contour for the numerical integration, but the response does little to address potential concerns about convergence or the accuracy of the obtained results, particularly for large values of t, for which the integrand can become highly oscillatory. Because the main takeaways of the paper rely on these numerics, more details about the method and its convergence would be important additions.